# Is One Layer Enough? Understanding Inference Dynamics in Tabular Foundation Models

**Amir Rezaei Balef** [1 2 3]  **Mykhailo Koshil** [1 2]  **Katharina Eggensperger** [1 2]

## Abstract

Transformer-based tabular foundation models (TFMs) dominate small to medium tabular predictive benchmark tasks, yet their inference mechanisms remain largely unexplored. We present the first large-scale mechanistic study of layerwise dynamics in 6 state-of-the-art tabular in-context learning models. We explore how predictions emerge across depth, identify distinct stages of inference and reveal latent-space dynamics that differ from those of language models. Our findings indicate substantial depthwise redundancy across multiple models, suggesting iterative refinement with overlapping computations during inference stages. Guided by these insights, we design a proof-of-concept, looped single-layer model that uses only 20% of the original model's parameters while achieving comparable performance. The code is available at https://github.com/amirbalef/is_one_layer_enough.

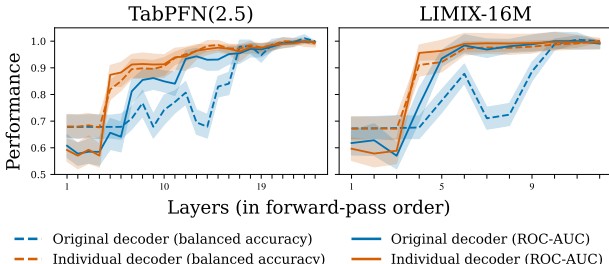

*Figure 1.* Individually trained decoders (**–**, **- -**) exhibit good performance early, showing that representations are descriptive but not aligned with the original decoder. For the original decoder, the sudden increase in ROC-AUC (**–**) and balanced accuracy (**- -**) at different layers suggests inference stages.

## 1. Introduction

Tabular foundation models (TFMs) have demonstrated that transformer-based architectures using in-context learning (ICL) can achieve state-of-the-art performance on small to medium-sized predictive tabular tasks (Erickson et al., 2025). Extending their flexibility and increasing performance, as well as exploring their potential for real-world industrial applications, form active areas of research. However, despite the ubiquity of tabular data, particularly in high-stakes settings such as healthcare and finance, the internal mechanisms of TFMs remain poorly understood.

Studying mechanism of inference by means of mechanistic explainability is necessary for two reasons. Firstly, it enables the identification of limitations in existing architectures and the detection of failures during model training and generalization, which are critical for advancing model development. Secondly, understanding how models use data for inference improves the predictability of model behavior in unseen settings, which is critical for reliable deployment (Sharkey et al., 2025).

It is unclear whether insights from architecturally similar large language models, i.e., LLMs (Gromov et al., 2025; Sun et al., 2025), transfer directly due to differences in inference strategies and training. Unlike most LLMs, prominent TFMs (e.g., *TabICL*, *TabPFN*) are encoder-only, smaller in size, do not perform auto-regressive inference, use attention between features, and are row-invariant, operating over sets of examples rather than sequences. Furthermore, LLMs are trained for next-token prediction on real-world data, making memorization of surface-level facts viable. This enables LLMs to derive the associations between "Paris" and "France" (Petroni et al., 2019) without such information being contained in the context. In contrast, TFMs are typically pre-trained to solve synthetic tabular tasks using ICL; therefore, it remains unclear whether and which inductive biases they learn or memorize.

These key differences motivate our study of the internal dynamics of tabular ICL models. More importantly, the smaller size and lower inference costs of TFMs enable large-scale studies that are often infeasible for LLMs. We present

---

[1]TU Dortmund University, Dortmund, Germany [2]Lamarr Institute for Machine Learning and Artificial Intelligence, Dortmund, Germany [3]University of Tübingen, Tübingen, Germany. Correspondence to: Amir Rezaei Balef <amir.balef@tu-dortmund.de>.

*Proceedings of the 43rd International Conference on Machine Learning*, Seoul, South Korea. PMLR 306, 2026. Copyright 2026 by the author(s).

an illustrative experiment that studies layer-wise performance, in which inference is halted after a given layer and the resulting hidden state is passed to a decoder to make a prediction. For LLMs, this evaluation typically applies the original decoder (i.e., the detokenizer) and is referred to as the "logit lens" (nostalgebraist, 2020). Belrose et al. (2023) showed that this can be brittle and proposed a "tuned lens", a learned affine transformation in each layer. For TFMs, we can efficiently continue pre-training the original decoder and adapt it to each layer. We show in Figure 1 that our "tabular tuned lens" leads to substantially different behaviour compared to the "logit lens" and that good performance is already possible in early layers. This raises the question of how many layers are actually needed to perform effective inference in TFMs, i.e., *Is one layer enough?*

To study this in detail, we provide the first large-scale mechanistic study of layer-wise dynamics over six state-of-the-art TFMs. Specifically, we carefully designed and adapted 6 experiments to study TFMs, inspired by mechanistic studies of LLMs, including structural interventions, ablations, and probing. Each experiment provides distinct and insightful observations; by combining these insights, we address the following two research questions.

**RQ1: How does inference unfold across depth in TFMs during prediction?** We identify where and how predictions are formed and assess the robustness of inference to layer ablations. We find that iterative inference and overlapping computations emerge across several state-of-the-art TFMs, which informs the design of more efficient architectures.

**RQ2: How do the inference dynamics of TFMs compare to those observed in LLMs?** We relate our findings to observations in LLM behavior, focusing on the emergence of inference stages and specialized layers (Lad et al., 2025). While it is evident that TFMs form block-layer structures like LLMs, they are more sensitive to layer swapping. Furthermore, although less pronounced, we observe similar stages of inference.

Finally, based on these findings, we design a proof-of-concept experiment showing that a single-layer TFM can achieve similar performance as a six-layer TFM, if we train it by *looping* layers. We conclude our study with three suggestions to explore for architectural improvements.

## 2. Related Work and Background

This section reviews prior work on understanding inference mechanisms, with a focus on representation analysis and layer-wise ablations in large language models. We also provide current views on how inference evolves in transformer models and related work on TFMs.

### 2.1. Methods for Studying Inference Mechanisms

Studying neural networks by examining how their internal parts, such as neurons, layers, or attention heads, interact to perform computations is commonly referred to as mechanistic interpretability (MI) (Olah et al., 2018; 2020; Zou et al., 2023; Sharkey et al., 2025). MI methods broadly fall into two categories. **Reverse engineering** approaches decompose a trained network into its constituent components and seek to infer the functional role each component plays in the overall computation (Wang et al., 2023). In contrast, **concept-based interpretability** starts from hypothesized concepts or variables and identifies network components that are critical to implement a concept (Elhage et al., 2021). Overall, MI aims to understand model behavior in order to detect failures or unsafe behavior, predict generalization and emergent abilities, modify internal mechanisms to align behavior with human objectives, improve training and inference, and extract latent knowledge to better model the world. While models for language and vision are already studied at the level of neurons (Sharkey et al., 2025), methods and insights for tabular models lag behind.

Given their architectural similarity to TFMs, we continue by briefly describing related methods for studying LLMs. First, studying **learned representations and data embeddings** can provide insights into the properties of hidden states. Several studies focus on the interpretability of LLMs using representation analysis (Dar et al., 2023; Song et al., 2025; Bronzini et al., 2025), for example, by using **probing classifiers** (Rogers et al., 2020) or projecting representations into the model's vocabulary space (nostalgebraist, 2020; Din et al., 2024; Belrose et al., 2023). Predictions of probes are then used to judge how much of the final prediction has already been formed at different stages of inference. Combining this with interventions, we can directly identify whether a representation is critical for certain predictions (Wang et al., 2023; Ghandeharioun et al., 2024).

Another line of work focuses on **layer-wise dynamics** of transformer models in LLMs. Skean et al. (2025) showed that intermediate layers often outperform final layers on downstream tasks. Similarly, Sun et al. (2025) analyzed representational alignment via cosine similarity and found that middle layers occupy a shared representation space, while early and late layers are more specialized. Through structural interventions such as skipping, reordering, or repeating layers, they further show that some middle layers can be removed without catastrophic failure, whereas repeating a single layer severely degrades performance. Complementing these findings, Lad et al. (2025) investigated the robustness of LLMs to structural interventions by deleting or swapping adjacent layers during inference. They found that early and final layers are the most sensitive, while middle layers are remarkably robust. This robustness is attributed

to the transformer's residual architecture. They further identified four depth-dependent stages of LLM inference: early layers perform detokenization, middle layers refine task- and entity-specific features, mid-to-late layers ensemble predictions, and final layers sharpen outputs by suppressing irrelevant features. These analyses are straightforward to conduct for TFMs; hence, they provide our starting point.

## 2.2. Two Views on Inference Dynamics

There are two non-competing hypotheses offering explanations for how inference evolves in artificial neural models, specifically transformers.

First, the **circuit hypothesis** presents a mechanistic view, proposing that individual model components perform specialized, modular roles along distinct computational pathways (Conmy et al., 2023). Evidence includes knowledge neurons and circuits (Dai et al., 2022; Yao et al., 2024), MLP units that suppress token repetition in copy-suppression mechanisms (McDougall et al., 2023), and task-general or 'universal' units (Gurnee et al., 2024; Voita et al., 2024).

Second, the **iterative inference hypothesis** proposes that the residual stream refines representations iteratively rather than learning entirely new ones (Greff et al., 2017; Jastrzeb-ski et al., 2018), for example, through skip connections in ResNet architectures. Similarly, in transformer models, the residual stream accumulates each layer's contribution via linear projections, progressively refining representations in a flexible, high-dimensional space (Elhage et al., 2021). This suggests that each layer incrementally updates the residual stream to improve the prediction (Geva et al., 2022; Belrose et al., 2023). Iterative inference is additionally supported by self-repair (Wang et al., 2023; McGrath et al., 2023; Rushing & Nanda, 2024), showing that later layers correct or mitigate errors of earlier layers. This behaviour arises if multiple layers perform similar or overlapping computations (Rushing & Nanda, 2024). As subsequent layers can compensate for removing earlier layers, we must consider this for ablation-based interpretability.

Importantly, recognizing that models iteratively refine their internal representations allows us to make more effective use of a given model size, enabling smaller models without sacrificing performance. A popular example is a **looped transformer** where repeating (looping) transformer blocks (instead of training deeper models) improves performance (Dehghani et al., 2019; Gong et al., 2025; Zhu et al., 2025). We build on this idea and study the role of overlapping computations and iterative inference in TFMs to explore the potential of recurrent model components to improve efficiency and performance.

## 2.3. Tabular Foundation Models

TFMs are an emerging type of transformer-based models, pre-trained to solve supervised learning tasks via in-context learning (ICL). Specifically, provided with a support set of (labeled) training samples, the model has learned to approximate Bayesian inference for the (unlabeled) query samples. This is referred to as ICL, since the model performs inference without updating weights. The current generation of TFMs is commonly pre-trained on synthetic tasks generated by a prior distribution, i.e., they are based on a prior-data-fitted network (Müller et al., 2022).

*TabPFN(v1)* (Hollmann et al., 2023) was the first widely used model operating in this fashion on tabular data, with a vanilla transformer backbone. *TabPFN(v2)* (Hollmann et al., 2025) improves by adding an attention mechanism within tokens, in addition to cross-token attention. *TabICL* (Qu et al., 2024) shares a similar backbone with *TabPFN(v2)*, but additionally introduces a transformer-based compression that efficiently transforms rows into semantically rich embeddings. Specifically, *TabICL* employs a two-stage architecture: first, it compresses the data, and then uses these embeddings to make predictions. LimiXTeam (2025) further extends this line of work by introducing two TFMs, *LimiX-2M* and *LimiX-16M*, which offer enhanced handling of missing values and support retrieval-based ensemble methods. More recently, *TabPFN(2.5)* (Grinsztajn et al., 2025) scales to datasets with up to $50\,000$ rows and $2\,000$ features, achieving state-of-the-art performance. Notably, *TabICL* and *TabPFN(v1)* are the only open-source TFMs with access to training and prior-data generation.

Prior work on understanding how these TFMs operate is largely limited to the *TabPFN* model family. Nagler (2023) analyzed how PFNs approximate predictive posteriors, providing a statistically grounded framework to understand how their ICL mechanism works with a focus on the bias-variance tradeoff and the need for localization. McCarter (2024) studied the behavior of *TabPFN(v1)* and *TabPFN(v2)* on out-of-distribution tasks to assess how well the model generalizes on data that is not described by its prior. Zheng et al. (2025) used concepts from signal reconstruction and frequency response analysis to investigate the inductive biases of *TabPFN(v2)*, demonstrating that *TabPFN(v2)* can dynamically adjust its frequency capacity to the number of support samples provided. Ye et al. (2025) showed that *TabPFN(v2)* learns highly predictive features and can be used to embed tabular data for downstream tasks. We aim to study general patterns in TFMs, going beyond focusing on individual models.

## 3. Empirical Experiments

Here, we present our main empirical results and start by describing the general setup and models considered.

**Models.** In the experiments we study two state-of-the-art open-source tabular ICL models, *TabPFN(v1)* (Hollmann et al., 2023) and *TabICL* (Qu et al., 2024) as well as four open-weight models, *TabPFN(v2)* (Hollmann et al., 2025), *TabPFN(2.5)* (Grinsztajn et al., 2025), *LimiX-2M* and *LimiX-16M* (LimiXTeam, 2025). We run all models in their default configurations with their standard data pre-processing; however, if applicable, we set the number of ensembles to 1. Additional details on the model architectures and their key differences are provided in Appendix A.2.

**Datasets.** For all experiments, we use binary classification tasks.[1] Specifically, we use a subset of 15 tasks from *TabArena*, selected to match the model constraints with $\leq 10,000$ samples and $\leq 100$ features (Erickson et al., 2025), as well as 34 tasks from on *PMLBmini* (Knauer et al., 2024), containing small datasets with $\leq 500$ samples. We report ROC-AUC, averaged across folds and repetitions (see Appendix A.5 for more details).

In the following, we describe our 6 experiments, each accompanied by its description, results, observations, and main takeaways highlighted in blue boxes. Each experiment studies one aspect of the inference process, and we order the experiments from low-level, comparing embeddings, to higher-level, ablating layers, as illustrated in Figure 2.

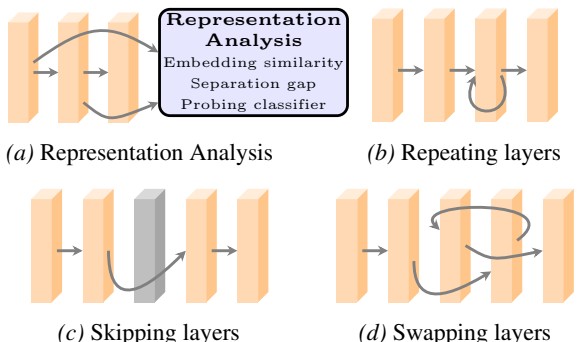

*(a)* Representation Analysis     *(b)* Repeating layers

*(c)* Skipping layers     *(d)* Swapping layers

*Figure 2.* Experiments analyzing the inference process of tabular ICL models.

①  **Embedding similarity.** First, we study the similarity of the representation space across layers. Following prior work (Sun et al., 2025; Lad et al., 2025), we examine both the averaged absolute cosine similarity and linear centered kernel alignment (CKA) (Kornblith et al., 2019) between the output embeddings of each layer (see Appendix A.1 for

---

[1]We adapt and run all experiments for both multiclass and regression tasks (see Appendix D and E). Overall, the results are consistent with and support our main findings.

more details).

High similarity between layers may suggest a shared representation space, whereas low similarity indicates that a layer significantly transforms the embedding space.

Results in Figure 3 show that adjacent layers generally exhibit a high similarity. In addition, all models, except *TabPFN(v1)* and *TabICL*, form clearly visible blocks of sequences of layers in which representation remains highly similar, suggesting small incremental updates to the embedding space. Moreover, while cosine similarity only reflects individual vector alignment, Davari et al. (2023) argued that CKA can be sensitive to certain affine transformations, e.g., non-isotropic scaling. This is evident in cases where high cosine similarity coexists with low CKA. One possible explanation is that specific attention heads stretch or scale certain embedding dimensions, producing high cosine similarity while reducing CKA. Per-benchmark results are reported in Appendix B.1.

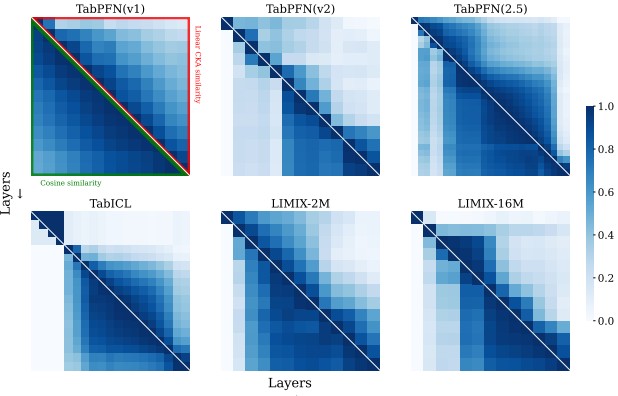

*Figure 3.* ①  **Embedding similarity** over different layers of the respective models (average over all datasets), upper triangular – linear CKA, lower triangular – cosine similarities.

> **Takeaway 1.** TFMs often form blocks in which the embeddings remain similar.

②  **Separation gap.** Unlike LLMs, TFMs have a fixed task, for example, classification. This allows us to track progress towards the goal of separating classes. To do so, we study the distance between representations of samples within the same class and between different classes across layers. We refer to the *separation gap* as the difference between the intra-class and inter-class distances. We randomly sample 100 pairs of data-points (within and across classes) and compute the pairwise cosine similarities between representations.[2] We compute the gap for support and query samples. Additionally, for models that use attention across

---

[2]To reduce noise in such high-dimensional spaces, we apply PCA while retaining 95% of the variance, and then compute distances in the projected space.

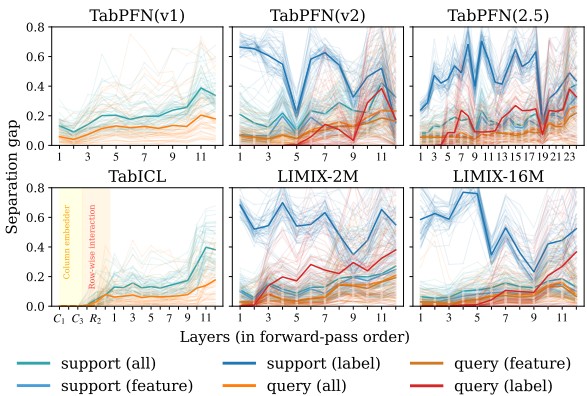

*Figure 4.* ② **Separation gap** (mean difference between inner-class and intra-class distances) across layers of the embedding network. Bold lines indicate the average across tasks, and thin lines represent results for individual datasets.

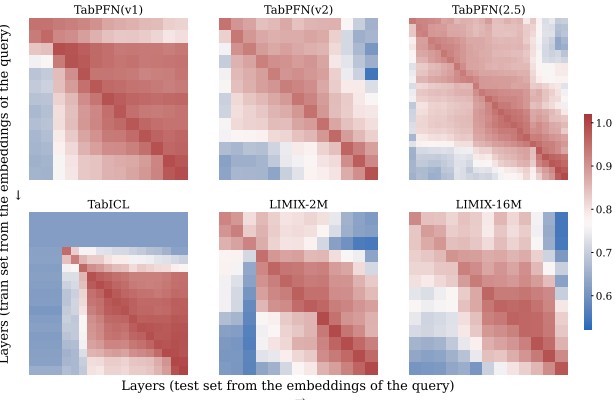

*Figure 5.* ③ **Probing classifiers** (logistic regression) normalized AUC trained on embeddings at different layers of the models.

features, we also compute the gap value for the (grouped) features and label. See Appendix B.2 for a formal definition and more details.

A large separation gap indicates highly discriminative features, making samples from different classes easier to distinguish. We study how this gap develops across layers to determine whether it gradually increases or undergoes sudden jumps. In Figure 4, we show that the separation gap for all models generally increases with depth, with some fluctuations.[3] Notably, the gap for the label embedding in the support set (**–**) stands out, as it already starts at a high value due to the presence of label information. Comparing the development of the gap of the label embedding (**–**) and the feature embedding (**–**), we observe a slight delay, suggesting that the models first focus on forming separable representations for the features and then for the label. Furthermore, we observe significant differences between models: *LimiX-2M* already shows a large gap in the early layers, whereas *TabPFN(v2)* shows a noticeable jump in layer 5.

> **Takeaway 2.** TFMs incrementally increase the distance between samples from different classes.

③ **Probing classifiers.** We use probing classifiers to measure how much information relevant to a downstream task is contained in embeddings from different model layers (Belinkov, 2022). We first extract query embeddings (as support embeddings inherently contain label information from the training data) from the hidden states of each layer during the forward pass. For this experiment, we extend the query set by including half of the original training set (excluded from the support set) to serve as training data for

the probing classifier, in our case logistic regression. We perform linear probing across layers by evaluating classifiers on representations from the same and different layers. Additional details and alternative probing methods are provided in Appendix B.3.

We use the performance of this probe as an approximation to the mutual information between the embedding and the quantity of interest (our label).

The results in Figure 5 show a consistent, though model-dependent, pattern: a probe trained on layer $(i)$ generalizes better to embeddings from later layers $(j > i)$ than the reverse, indicating that later layers retain information from earlier layers while also encoding new features that are not present in lower layers.

> **Takeaway 3.** Each layer cumulatively enriches the representation by adding new features while preserving previous ones.

④ **"Tabular" Logit Lens.**

For the next experiment, we use our adapted version of the popular "logit lens" method (nostalgebraist, 2020). As shown in the introduction, we can not rely on the original final decoder and use individual decoders per layer (see also irregular entropy patterns across layers in Figure B.11).

Following Küken et al. (2025), we continue pre-training our final decoder individually for each layer on synthetic datasets generated by *TabICL* priors (details are provided in Appendix A.6). Then, after each layer, we pass the embeddings to its individual decoders.

We study the performance of individual decoders as a proxy to measure whether the features crucial for a prediction (for a given task) were already formed at a certain layer.

Results in Figure 6 show that high performance can already be achieved in early layers for all models. Further-

---

[3]We note that fluctuations may arise from nonlinearities in the representations, which our metric cannot fully capture.

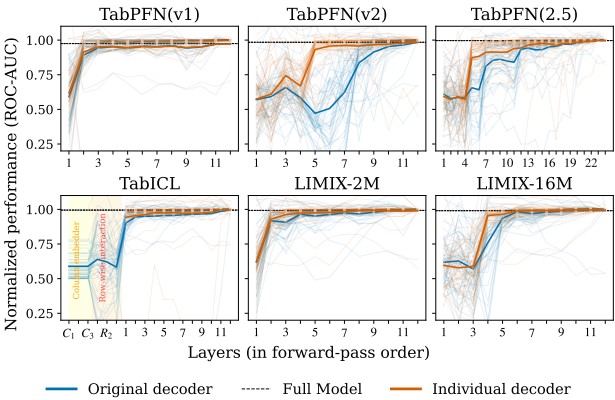

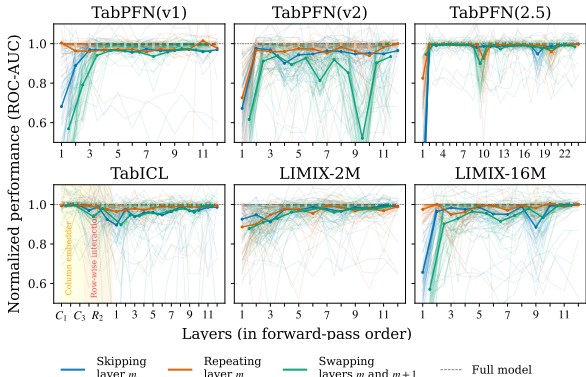

*Figure 6.* ④ **"Tabular" Logit Lens** on the respective tabular foundation model.

*Figure 7.* ⑤ **Layer ablation** effect on the performance of model.

more, performance increases abruptly for all models, and some models show further incremental improvements (e.g., *TabPFN(2.5)*). Notably, using individual decoders yields faster and more reliable predictions than applying the original decoder as a logit lens.

> **Takeaway 4.** Representations are already formed for a reliable prediction in the early layers, but not necessarily aligned with the original decoder.

⑤ **Layer ablation.** We perform layer ablation and reconfiguration experiments to study the role and contribution of individual layers. We manipulate the execution order of layers by skipping, repeating, and swapping them, and measure the resulting performance of the forward pass (see Appendix B.5 for more details and results).

Skipping a layer indicates the layer contribution. By repeating a layer, we test whether the model has learned to perform the same refinement iteratively across layers. In such a case, repeating the layer would be expected to improve performance. Finally, swapping layers tests sequential representational alignment. If layers are not aligned sequentially, the order of execution can be changed without loss of performance. Figure 7 shows that skipping an early layer leads to the largest performance degradation, whereas skipping middle and later layers results in little to no performance drop. This indicates that layers contribute differently to forming the final predictions. Repeating certain layers results in slight performance improvements for models such as *LimiX-16M* and *TabPFN(v1)*, supporting the idea of iterative refinement. Models are generally sensitive to layer swapping. One reason could be that some layers learn specialized, co-adapted representations that swapping layers disrupts this feature hierarchy, leading to degraded performance. Another explanation is that swapping layers simultaneously applies interventions to two layers, thereby making recovery more difficult.

> **Takeaway 5.** Early layers contribute the most, while later layers perform iterative refinement of the representation.

⑥ **Self-repair.** Here, we study whether TFMs exhibit self-repair mechanisms and, thus, whether layers perform similar or overlapping computations. As observed, TFMs are generally robust to ablating layers. However, it is unclear whether this robustness arises from self-repair or from layer redundancy, as we have measured performance only at the final layer. We use our "tabular logit lens" to measure the performance of all subsequent layers after skipping a layer (see Appendix B.6 for more results and details). If the TFM can recover from dropping a layer, it has learned to self-repair, and layers overlap in functionality.

Results in Figure 8 show that interventions in early layers cannot be recovered and, thus, implement unique functionality; however, self-repair is clearly visible in middle and later layers, especially for *TabPFN(v2)*.

> **Takeaway 6.** Self-repair generally occurs after layer ablations, except for the first layer.

## 4. Discussion

We will now discuss the results in the context of our research questions and draw connections to prior work.

**RQ1: How does inference unfold across depth in TFMs during prediction?** Our experiments show that the separation gap increases and each successive layer increases the distance between feature embeddings of different classes. Additionally, probing experiments show that each layer introduces increasingly stronger and more informative features. These results strongly support the iterative inference hypothesis, as they show that each layer's contribution is incremental. In the embedding similarity experiment, we observe that a block structure emerges in the largest models (e.g., *TabPFN(2.5)* and *LimiX-16M*). Within each block,

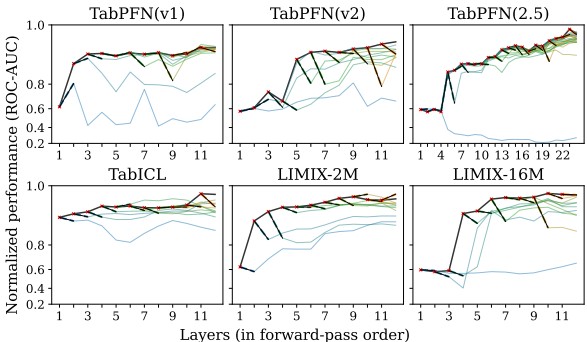

*Figure 8.* ⑥ **Self-repair** analysis under layer skipping. The Tabular Logit Lens measures model performance at each layer (intermediate performance). The solid black line shows intermediate performance without intervention. Colored lines, from blue (early) to orange (late), show intermediate performance after layer ablations, with cross markers indicating skipped layers. Dashed lines connect the first layer after a skipped layer to its original performance; a drop followed by recovery indicates self-repair.

the embedding changes gradually, whereas it changes more drastically between blocks. Furthermore, we observed that while most layers are robust to intervention (see layer ablation and self-repair experiments), a small number, especially the first layer in most models, are highly sensitive to being skipped, suggesting specialization.

Interestingly, two models showed a distinct pattern: *TabICL* and *LimiX-2M* exhibit more robustness to skipping early layers compared to other models, as evidenced in Figure 7. *TabICL*'s prediction module operates directly on already processed embeddings from the column embedder and row-wise interaction module, and *LimiX-2M* uses an RBF-kernel preprocessing step. This results in strong features being produced upfront; consequently, the models rely less on the first transformer layer. Since *LimiX-16M* does not employ the same preprocessing and is more dependent on the first few layers, we conclude that the primary functional role of the early layers is to map the representations from the input encoder to representations that are suitable for the residual stream operations.

**RQ2: How do the inference dynamics of TFMs compare to those observed in LLMs?** Beginning with differences, most notably and in contrast to LLMs, redundancy plays a larger role in the middle layers, as suggested by the relatively stable, high performance observed in the early-exit experiment shown in Figure 6. This statement is further supported by the stark decreasing entropy in early layers (as shown in Figure B.11 in the Appendix), which means that a confident prediction is formed early on.

Additionally, TFMs are considerably more sensitive to layer swapping than LLMs (Sun et al., 2025), with this sensitivity being especially pronounced in the *TabPFN(v2)* model. While the final layer in both model types increases the sepa-

ration gap, its functional role differs: in LLMs, removing or disrupting the last layer typically leads to a noticeable performance drop, likely because it performs residual sharpening (i.e., amplifying and consolidating already-formed representations to ensure confident token-level predictions). In TFMs, by contrast, the last layer seems less important; although it improves class separation, its removal or perturbation does not degrade performance to the same extent, suggesting that TFMs rely less on late-stage residual refinement and more on earlier layers for forming decisive representations.

However, we also find clear commonality in the inference dynamics of TFMs and LLMs. Firstly, early layers are crucial, and their ablation drastically reduces performance. Secondly, we observe the emergence of blocks of layers operating on similar embeddings.

The presence of blocks of layers with high similarity suggests the existence of stages of inference (see related work in Section 2). However, we argue that these stages are distributed differently across the layers in TFMs. To reflect this, we adjust the names of the stages accordingly, see Appendix C for a detailed comparison. Based on Figure 6, the first stage LATENT MAPPING is an extension of the input encoder, where representations are transformed into enriched feature representations for subsequent layers. For models with a more advanced encoding, such as *LimiX-2M* and *TabICL*, this stage is less pronounced. The second stage is FEATURE ENGINEERING AND LABELING, during which all models show rapid improvements in individual decoder performance. During this stage, features of the same class move closer together, while features of different classes gradually separate, and labels are formed iteratively. The third stage, PREDICTION ENSEMBLING, is less pronounced and only clearly observed in *TabPFN(v2)* and *TabPFN(2.5)*. It occurs when the early-exit performance of the individual decoder (–) has already converged, but the original decoder (–) continues to improve. In this stage, representations are transformed to better align with the original decoder. The final stage, PREDICTION CALIBRATION, happens when both decoders have similarly high performance, but the balanced accuracy exhibits a noticeable jump, and the output entropy continues to change (see Figures B.12 and B.11). Notably, these stages can overlap due to computations that overlap and redundancy across layers, as well as the self-repair mechanism.

## 5. Is One Layer Enough?

Our experiments show that most non-early layers appear redundant in tabular logit lens and layer-ablation experiments; however, we find that these layers still influence prediction quality, i.e., they are important for performance but might overlap in computation. This suggests that fewer, but re-

peated layers, could achieve similar performance, which connects to our initial question: Is one layer enough? In this concluding experiment, we illustrate the impact of this finding.

Specifically, we leverage the open-source *nanoTabPFN* (Pfefferle et al., 2025) codebase providing an architecture similar to *TabPFN(v2)* and pre-train three models using the *TabICL* prior codebase: (1) the original implementation containing six layers without any modification (*nanoTabPFN$_{6l}$*), (2) a single-layer transformer (*nanoTabPFN$_{1l}$*) and (3) a single-layer transformer where we repeat the single layer up to six times during trained and inference (*nanoTabPFN$_{looped}$*) (see Appendix A.3 for details). We highlight that *nanoTabPFN$_{looped}$* matches the computational complexity of *nanoTabPFN$_{6l}$* while retaining only the number of parameters of *nanoTabPFN$_{1l}$*. This controlled comparison suggests that the observed performance gains are not merely due to parameter count but rather to the iterative (looping) mechanism.

We compare these models, including original *TabPFN(v1)* and *TabPFN(2.5)* on the *PMLBmini* and *TabArena* benchmarks and show results in Figure 9. We first observe that the single-layer model (–) clearly compares worse than all other models, indicating that a single layer alone is insufficient. However, more importantly, *nanoTabPFN$_{looped}$* (–) performs almost identically to the six-layer *nanoTabPFN* (–); the model learns to iteratively refine its predictions, reusing a single layer. We also observe that performance is similar across the number of layers and loops. This supports the view that depth in TFMs primarily facilitates iterative computation rather than learning fundamentally distinct transformations at each layer, and that comparable predictive performance can be achieved through recurrent reuse of a single transformer block. Notably, one advantage of *nanoTabPFN$_{looped}$* is that it can produce immediate predictions without requiring pretraining of individual decoders. However, we note that our experiments were conducted at a small scale, and there remains a performance gap compared to state-of-the-art models such as *TabPFN(2.5)*.

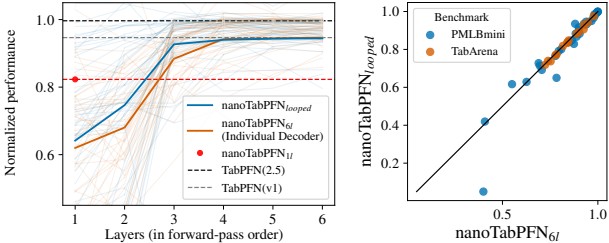

*Figure 9.* Performance comparison between *nanoTabPFN$_{6l}$*, *nanoTabPFN$_{1l}$*, and *nanoTabPFN$_{looped}$*. Repeating a single transformer block recovers performance comparable to the full-depth model.

## 6. Conclusion

In this work, we investigate open questions regarding the mechanisms inside TFMs and how they solve predictive tasks. We found that although the layer dynamics in the TFMs we studied differ from those in LLMs, iterative refinement plays a major role in how inference unfolds, layers form blocks with overlapping functionality, and different inference stages emerge. Specifically, we can derive insights that open new directions for a principled improvement of TFMs:

- Using a strong encoder leads to more robust inference dynamics and forming highly descriptive features in early layers. Our results suggest that the compression stage of *TabICL* and the encoding used in *LimiX-2M* yield a richer embedding of raw data.

- The near-complete recovery of original performance after layer ablation indicates depth-wise redundancy. Our results show that TFMs can reliably recover from the removal of middle and late layers, suggesting that less capacity (i.e., shallower models) is required to achieve this performance.

- The emergence of layer-wise blocks of highly similar representation reveals further opportunities for post-hoc model compression, such as merging layers or replacing multiple blocks with a recurrently applied shared layer to improve parameter efficiency (McLeish et al., 2025).

**Limitations.** Our work has several limitations. First, we did not study in detail when effective depth becomes necessary, and leave a more systematic investigation of this question, e.g., using controlled benchmarks with varying notions of tabular task complexity, for future work. For Tabular Logit Lens, we use the open-source priors from *TabICL*, which may be suboptimal for models that employ more expressive prior designs. For the looped transformer experiments, we use the smaller *nanoTabPFN* architecture. Although the results do not directly extrapolate to larger architectures without further experiments, they are encouraging and suggest that this approach can effectively scale to larger models such as *TabPFN(2.5)*. Finally, we evaluated the models without ensembling and on only two benchmark suites, studying mostly average performance; the results could be impacted by different benchmarks and ensembling.

**Future Work.** We see several promising steps to build on our work: (1) extending our experimental setup to study TFMs at the neuronal and circuit level to gain deeper mechanistic insights, (2) using our methods to study the impact of design choices like the prior design and pre-training setup and (3) study whether to LLM-based models for predictive tabular tasks Hegselmann et al. (2023); Gardner et al. (2024)

exhibit similar behaviour. Such studies could reveal whether observed patterns are consistent and help identify potential weaknesses in the model architecture, learned biases, or the training data. Finally, we aim to further investigate recurrent models, such as looped transformers at larger scales, which offer benefits such as anytime predictions, parameter efficiency, and adaptive computation, which represent a particularly promising direction for future research.

## Acknowledgments

This research has been funded by the Federal Ministry of Research, Technology and Space of Germany and the state of North Rhine-Westphalia as part of the Lamarr Institute for Machine Learning and Artificial Intelligence. Additionally, part of this research utilized compute resources at the Tübingen Machine Learning Cloud, DFG FKZ INST 37/1057-1 FUGG. A. Balef and M. Koshil also thank the International Max Planck Research School for Intelligent Systems (IMPRS-IS).

## Impact Statement

This paper presents work whose goal is to advance the field of Machine Learning. There are many potential societal consequences of our work, none of which we feel must be specifically highlighted here.

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

# A. Experimental setup details

## A.1. Metrics

**Linear centered kernel alignment (CKA)** is a metric (Kornblith et al., 2019) used to evaluate the similarity between representations. Given two column-centered feature matrices $X \in \mathbb{R}^{n \times d_1}$ and $Y \in \mathbb{R}^{n \times d_2}$, linear CKA is defined as

$$\text{CKA}(X, Y) = \frac{\|X^\top Y\|_F^2}{\|X^\top X\|_F \cdot \|Y^\top Y\|_F + \varepsilon}, \tag{1}$$

where $\|\cdot\|_F$ denotes the Frobenius norm and $\varepsilon$ is a small constant for numerical stability. This metric quantifies the similarity between two representations while being invariant to isotropic scaling.

**Cosine similarity** is a metric used to measure the similarity between two vectors. Given two vectors $x, y \in \mathbb{R}^d$, cosine similarity is defined as

$$\text{CosSim}(x, y) = \frac{x \cdot y}{\|x\|_2 \, \|y\|_2}, \tag{2}$$

where $\|\cdot\|_2$ denotes the $\ell_2$ norm. This metric quantifies the angular similarity between two vectors, independent of their magnitudes. In our experiments, we report the average cosine similarities.

## A.2. Architectures

In this section, we provide additional details on the tabular in-context learning (ICL) models used in our experiments. We focus on architectural differences, representation strategies, and key design choices.

*TabPFN(v1)* is the first version of the Tabular Prior-Data Fitted Network (TabPFN) designed for in-context learning on tabular data (Hollmann et al., 2023). Unlike later TabPFN variants, *TabPFN(v1)* employs a single embedding per feature and row, without additional cross-feature attention mechanisms as seen in Table A.1.

*Table A.1.* Layers and Parameters in *TabPFN(v1)*

| Layer Name | Description | Num Parameters |
|---|---|---|
| *TabPFN(v1)* | - | 25,821,706 |
| Input encoder | Linear + Linear | 51,712 + 1,024 |
| Transformer blocks | 12 layers | 25,233,408 |
| Each block | - | 2,102,784 |
| MultiheadAttention | - | 1,050,624 |
| LayerNorm1 | Post-LN | 1,024 |
| MLP | Linear + Linear | 525,312 + 524,800 |
| LayerNorm2 | Post-LN | 1,024 |
| Decoder | Linear + Linear | 525,312 + 10,250 |

*TabPFN(v2)* is an improved version of the original TabPFN (Hollmann et al., 2025), designed to increase model capacity and generalization for in-context learning on tabular data. Compared to *TabPFN(v1)*, it includes additional transformer layers and attention heads as seen in Table A.2, enabling richer representations and more complex interactions between input features.

*Table A.2.* Layers and Parameters in *TabPFN(v2)*

| Layer Name | Description | Num Parameters |
|---|---|---|
| *TabPFN(v2)* | PerFeatureTransformer | 7,244,554 |
| **Encoder** | SequentialEncoder | - |
| Feature | Linear | 768 |
| Label | Linear | 576 |
| Positional embedding | Linear | 9,408 |
| **Transformer Blocks** | 12 layers | 7,077,888 |
| Each block | - | 589,824 |
| MultiheadAttention | Attention Between Features | 147,456 |
| LayerNorm | Post-LN | 0 |
| MultiheadAttention | Attention Between Items | 147,456 |
| LayerNorm | Post-LN | 0 |
| MLP | Linear + Linear | 147,456 + 147,456 |
| LayerNorm | Post-LN | 0 |
| Decoder | Linear + Linear | 148,224 + 7,690 |

**TabPFN(2.5)** (Grinsztajn et al., 2025) is the latest TabPFN variant, roughly doubling the number of layers compared to *TabPFN(v2)* (Table A.2). This increase in depth allows the model to capture more complex feature interactions and improves generalization while maintaining in-context learning capabilities.

*Table A.3.* Layers and Parameters in *TabPFN(2.5)*

| Layer Name | Description | Num Parameters |
|---|---|---|
| *TabPFN(2.5)* | PerFeatureTransformer | 10,718,218 |
| **Encoder** | - | - |
| Feature | Linear | 1,152 |
| Label | Linear | 576 |
| Positional embedding | Linear | 9,408 |
| **Transformer Blocks** | 24 layers | 10,616,832 |
| Each block | - | 442,368 |
| MultiheadAttention | Attention Between Features | 147,456 |
| LayerNorm | Post-LN | 0 |
| MultiheadAttention | Attention Between Items | 147,456 |
| LayerNorm | Post-LN | 0 |
| MLP | Linear + Linear | 73,728 + 73,728 |
| LayerNorm | Post-LN | 0 |
| Decoder | Linear + Linear | 74,112 + 3,850 |

**TabICL** (Qu et al., 2024) is a transformer-based in-context learner for tabular data that uses feature-wise embeddings, cross-feature attention, and learned positional encodings; from an in-context prediction perspective, it operates similarly to *TabPFN(v1)*, without additional cross-feature mechanisms, as summarized in Table A.4.

*Table A.4.* Layers and Parameters in *TabICL*

| Layer Name | Type | Num Parameters |
|---|---|---|
| *TabICL* | | 27,051,666 |
| **Feature Encoder (Column Embedder)** | - | 877,824 |
|   Input Linear | Linear | 256 |
|   Transformer Blocks | 3 layers | 844,032 |
|     Each block | - | 281,344 |
|     LayerNorm | Pre-LN | 256 |
|     MultiheadAttention | Attention 1 | 66,048 + 16,512 |
|     LayerNorm | Pre-LN | 256 |
|     MLP | Linear layers | 33,024 + 32,896 |
|     LayerNorm | Pre-LN | 256 |
|     MultiheadAttention | Attention 2 | 132,480 + 66,048 |
|     LayerNorm | Pre-LN | 256 |
|     MLP | Linear layers | 33,024 + 32,896 |
|   Output linear | Linears | 16,512 + 256 |
| **Feature Encoder (Row Interaction)** | - | 398,216 |
|   Rotary Embedding | - | 8 |
|   Transformer Blocks | 3 layers | 397,440 |
|     Each block | - | 132,480 |
|     LayerNorm | Pre-LN | 256 |
|     MultiheadAttention | Attention | 66,048 + 16,512 |
|     LayerNorm | Pre-LN | 256 |
|     MLP | Linear layers | 33,024 + 32,896 |
|   Output | LayerNorm | 256 |
| **ICL predictor** | - | 25,775,626 |
|   Transformer Blocks | 12 layers | 25,233,408 |
|     Each block | - | 2,102,784 |
|     LayerNorm | Pre-LN | 1,024 |
|     MultiheadAttention | Attention | 1,050,624 + 262,656 |
|     LayerNorm | Pre-LN | 1,024 |
|     MLP | Linear layers | 525,312 + 524,800 |
| Decoder | Linear layers | 535,562 |

***LimiX-2M*** and ***LimiX-16M*** (LimiXTeam, 2025) are transformer-based tabular in-context learners that maintain separate embeddings per feature. *LimiX-2M* is the smaller variant (Table A.5), emphasizing feature encoding with an RBF kernel, while *LimiX-16M* is larger (Table A.6), enabling richer feature interactions. Despite these differences, both support in-context predictions similarly to *TabPFN(v2)*, with separate feature embeddings facilitating detailed analysis of feature-wise interactions.

*Table A.5.* Layers and Parameters in *LimiX-2M*

| Layer Name | Description | Num Parameters |
|---|---|---|
| *LimiX-2M* | PerFeatureTransformer | 2,377,837 |
| **Encoder** | - | 47,712 |
| Feature | - | 18,048 |
| Fusion Network | - | 28,224 |
| Label(cls) | - | 1,056 |
| Label(reg) | - | 288 |
| **Transformer Blocks** | 12 layers | 2,211,840 |
| Each block | - | 184,320 |
| MultiheadAttention | Attention Between Items | 73,728 |
| LayerNorm | Post-LN | 0 |
| MLP | Linear layers | 73,728 |
| LayerNorm | Post-LN | 0 |
| MultiheadAttention | Attention Between Features | 36,864 |
| LayerNorm | Post-LN () | 0 |
| Decoder(cls) | Linear layers | 41,098 |
| Decoder(reg) | Linear layers | 38,401 |
| Decoder(feature) | Linear layers | 38,786 |

*Table A.6.* Layers and Parameters in *LimiX-16M*

| Layer Name | Description | Num Parameters |
|---|---|---|
| *LimiX-16M* | PerFeatureTransformer | 16,526,413 |
| **Encoder** | - | 134,016 |
| Feature | - | 19,392 |
| Fusion Network | - | 111,744 |
| Label(cls) | - | 2,112 |
| Label(reg) | - | 576 |
| Positional embedding | Linear | 9,408 |
| **Transformer Blocks** | 12 layers | 15,925,248 |
| Each block | - | 1,327,104 |
| LayerNorm | Pre-LN | 0 |
| MultiheadAttention | Attention Between Features | 147,456 |
| LayerNorm | Pre-LN | 0 |
| MLP | Linear layers | 294,912 |
| LayerNorm | Pre-LN | 0 |
| MultiheadAttention | Attention Between Features | 147,456 |
| LayerNorm | Pre-LN | 0 |
| MLP | Linear layers | 294,912 |
| LayerNorm | Pre-LN | 0 |
| MultiheadAttention | Attention Between Items | 147,456 |
| LayerNorm | Pre-LN | 0 |
| MLP | Linear layers | 294,912 |
| Decoder(cls) | Linear layers | 155,914 |
| Decoder(reg) | Linear layers | 150,529 |
| Decoder(feature) | Linear layers | 151,298 |

### A.3. *nanoTabPFN*

**nanoTabPFN**[4] is a simplified and lightweight implementation of the *TabPFN(v2)* architecture (Pfefferle et al., 2025). The architecture, shown in Table A.8, consists of six transformer blocks. For *nanoTabPFN$_{looped}$*, we use the same architecture but with a single transformer block, which is looped six times during the forward pass. This modification reduces the total number of parameters by 20%.

*Table A.7.* Layers and Parameters in *nanoTabPFN*

| Layer Name | Description | Num Parameters |
|---|---|---|
| *nanoTabPFN* | PerFeatureTransformer | 3,717,514 |
| **Encoder** | SequentialEncoder | - |
| Feature | Linear | 384 |
| Label | Linear | 384 |
| **Transformer Blocks** | 6 layers | 3,560,832 |
| Each block | - | 593,472 |
| MultiheadAttention | Attention Between Features | 185,280 |
| LayerNorm | Post-LN | 384 |
| MultiheadAttention | Attention Between Items | 185,280 |
| LayerNorm | Post-LN | 384 |
| MLP | Linear + Linear | 148,224 + 147,648 |
| LayerNorm | Post-LN | 384 |
| Decoder | Linear + Linear | 148,224 + 7,690 |

*Table A.8.* Layers and Parameters in *nanoTabPFN$_{looped}$*

| Layer Name | Description | Num Parameters |
|---|---|---|
| *nanoTabPFN$_{looped}$* | PerFeatureTransformer | 750,154 |
| **Encoder** | SequentialEncoder | - |
| Feature | Linear | 384 |
| Label | Linear | 384 |
| **Single Transformer Block** | - | 593,472 |
| MultiheadAttention | Attention Between Features | 185,280 |
| LayerNorm | Post-LN | 384 |
| MultiheadAttention | Attention Between Items | 185,280 |
| LayerNorm | Post-LN | 384 |
| MLP | Linear + Linear | 148,224 + 147,648 |
| LayerNorm | Post-LN | 384 |
| Decoder | Linear + Linear | 148,224 + 7,690 |

**Training.** We train all *nanoTabPFN* models using the hyperparameters listed in Table A.9. We report the pretraining cost on an NVIDIA A100 GPU in Table A.10.

---

[4]We use the codebase provided at `https://github.com/automl/TFM-Playground/`.

*Table A.9.* Training configuration and default hyperparameters.

| Hyperparameter | Value |
|---|---|
| Training steps | 10 000 |
| Batch size | 512 |
| Micro-batch size | 4 |
| Learning rate ($\eta$) | $1 \times 10^{-4}$ |
| Scheduler | Cosine warmup |
| Warmup steps | 2 000 |
| Gradient clipping | 1.0 |
| Weight decay | 0.0 |
| Cosine cycles | 1 |
| Cosine amplitude decay | 1.0 |
| Cosine final learning rate | 0.0 |
| Polynomial final learning rate | $1 \times 10^{-7}$ |
| Polynomial decay power | 1.0 |

*Table A.10.* Pretraining Runtime (hours)

| Model | Training time (hours) |
|---|---|
| $nanoTabPFN_{1l}$ | 11.9 |
| $nanoTabPFN_{6l}$ | 62.3 |
| $nanoTabPFN_{looped}$ | 68.8 |

## A.4. Reproducibility

**Code.** The code will be made publicly available upon acceptance.

**Compute cost.** All experiments were run on a compute cluster with 16 CPU cores per node and NVIDIA A100 GPU nodes. The total CPU and GPU hours for each benchmark and model are reported in Table A.11. GPU hours account only for active GPU utilization; the corresponding wall-clock GPU runtime is approximately given by the reported CPU hours divided by 16.

*Table A.11.* Total CPU and GPU hours per benchmark and model

| benchmark | model | CPU hours | GPU hours |
|---|---|---|---|
| *PMLBmini* | *TabPFN(v1)* | 5942.06 | 26.75 |
| *PMLBmini* | *TabPFN(v2)* | 6579.17 | 2.78 |
| *PMLBmini* | *TabPFN(2.5)* | 25329.90 | 5.64 |
| *PMLBmini* | *TabICL* | 15664.14 | 4.27 |
| *PMLBmini* | *LimiX-2M* | 5685.21 | 4.06 |
| *PMLBmini* | *LimiX-16M* | 9848.21 | 4.71 |
| *TabArena* | *TabPFN(v1)* | 18236.49 | 200.11 |
| *TabArena* | *TabPFN(v2)* | 25129.97 | 47.05 |
| *TabArena* | *TabPFN(2.5)* | 74645.58 | 65.21 |
| *TabArena* | *TabICL* | 38908.78 | 44.72 |
| *TabArena* | *LimiX-2M* | 26737.44 | 88.37 |
| *TabArena* | *LimiX-16M* | 39022.71 | 121.88 |

## A.5. Benchmarks

We use the *TabArena* and *PMLBmini* benchmarks to evaluate our experiments. For *TabArena*, we consider only binary classification tasks, in line with the limitations of the models (at most 10,000 samples and 100 features), resulting in 15 distinct tasks. Tables A.12 and A.13 list all datasets used in our experiments. For each dataset, we report the OpenML task ID (Bischl et al., 2025), dataset name, number of samples, number of features, and number of categorical features.

*Table A.12.* Datasets in *TabArena*.

| index | task id | dataset name | #samples | #features | #categorical features | #folds | #repetitions |
|---|---|---|---|---|---|---|---|
| 1 | 363619 | Bank-Customer-Churn | 10000 | 11 | 5 | 3 | 3 |
| 2 | 363624 | coil2000-insurance-policies | 9822 | 86 | 4 | 3 | 3 |
| 3 | 363623 | churn | 5000 | 20 | 5 | 3 | 3 |
| 4 | 363706 | taiwanese-bankruptcy-prediction | 6819 | 95 | 1 | 3 | 3 |
| 5 | 363689 | NATICUSdroid | 7491 | 87 | 87 | 3 | 3 |
| 6 | 363694 | polish-companies-bankruptcy | 5910 | 65 | 1 | 3 | 3 |
| 7 | 363700 | seismic-bumps | 2584 | 16 | 4 | 3 | 3 |
| 8 | 363674 | hazelnut-spread-contaminant-detection | 2400 | 31 | 1 | 3 | 10 |
| 9 | 363682 | Is-this-a-good-customer | 1723 | 14 | 9 | 3 | 10 |
| 10 | 363629 | diabetes | 768 | 9 | 1 | 3 | 10 |
| 11 | 363671 | Fitness-Club | 1500 | 7 | 4 | 3 | 10 |
| 12 | 363626 | credit-g | 1000 | 21 | 14 | 3 | 10 |
| 13 | 363684 | Marketing-Campaign | 2240 | 26 | 9 | 3 | 10 |
| 14 | 363621 | blood-transfusion-service-center | 748 | 5 | 1 | 3 | 10 |
| 15 | 363696 | qsar-biodeg | 1054 | 42 | 6 | 3 | 10 |

*Table A.13.* Datasets in *PMLBmini*.

| index | task id | dataset name | #samples | #features | #categorical features | #folds | #repetitions |
|---|---|---|---|---|---|---|---|
| 1 | 13 | breast-cancer | 286 | 10 | 10 | 10 | 1 |
| 2 | 27 | colic | 368 | 23 | 16 | 10 | 1 |
| 3 | 39 | sonar | 208 | 61 | 1 | 10 | 1 |
| 4 | 42 | haberman | 306 | 4 | 2 | 10 | 1 |
| 5 | 52 | heart-statlog | 270 | 14 | 1 | 10 | 1 |
| 6 | 54 | hepatitis | 155 | 20 | 14 | 10 | 1 |
| 7 | 55 | vote | 435 | 17 | 17 | 10 | 1 |
| 8 | 57 | ionosphere | 351 | 35 | 1 | 10 | 1 |
| 9 | 3495 | SPECT | 267 | 23 | 23 | 10 | 1 |
| 10 | 3496 | SPECTF | 349 | 45 | 1 | 10 | 1 |
| 11 | 3503 | aids | 50 | 5 | 3 | 10 | 1 |
| 12 | 3538 | analcatdata-boxing2 | 132 | 4 | 4 | 10 | 1 |
| 13 | 3539 | prnn-crabs | 200 | 8 | 2 | 10 | 1 |
| 14 | 3540 | analcatdata-boxing1 | 120 | 4 | 4 | 10 | 1 |
| 15 | 3542 | analcatdata-lawsuit | 264 | 5 | 2 | 10 | 1 |
| 16 | 3543 | irish | 500 | 6 | 4 | 10 | 1 |
| 17 | 3550 | analcatdata-asbestos | 83 | 4 | 3 | 10 | 1 |
| 18 | 3552 | analcatdata-creditscore | 100 | 7 | 4 | 10 | 1 |
| 19 | 3554 | backache | 180 | 32 | 27 | 10 | 1 |
| 20 | 3555 | prnn-synth | 250 | 3 | 1 | 10 | 1 |
| 21 | 3556 | analcatdata-cyyoung8092 | 97 | 11 | 4 | 10 | 1 |
| 22 | 3558 | analcatdata-japansolvent | 52 | 9 | 2 | 10 | 1 |
| 23 | 3562 | lupus | 87 | 4 | 1 | 10 | 1 |
| 24 | 3565 | analcatdata-bankruptcy | 50 | 6 | 2 | 10 | 1 |
| 25 | 3568 | analcatdata-cyyoung9302 | 92 | 10 | 5 | 10 | 1 |
| 26 | 3570 | biomed | 209 | 9 | 2 | 10 | 1 |
| 27 | 3819 | molecular-biology-promoters | 106 | 58 | 59 | 10 | 1 |
| 28 | 146188 | analcatdata-fraud | 42 | 12 | 12 | 10 | 1 |
| 29 | 146196 | corral | 160 | 7 | 7 | 10 | 1 |
| 30 | 146208 | mux6 | 128 | 7 | 7 | 10 | 1 |
| 31 | 146210 | postoperative-patient-data | 88 | 9 | 9 | 10 | 1 |
| 32 | 146236 | cleve | 303 | 14 | 9 | 10 | 1 |
| 33 | 146240 | parity5 | 32 | 6 | 6 | 10 | 1 |
| 34 | 3722 | hungarian | 294 | 14 | 8 | 10 | 1 |

## A.6. Tabular Logit Lens

Similar to (Küken et al., 2025), we pretrain individual decoders for each tabular foundation model using *TabICL* priors (see Appendix A.7). Notably, the individual decoders share the same architecture as the original decoder. During pretraining, all model parameters are frozen, and only the decoders are updated, using the settings shown in Table A.14. Table A.15 reports runtime of pretraining all decoders for each model on a single NVIDIA A100 GPU. Additional results are reported in Appendix B.4.

*Table A.14.* Training Parameters

| Parameter | Value |
|---|---|
| Epochs | 200 |
| Batch size | 8 |
| #Steps/Epoch | 512 |
| Learning rate | 3e-5 |

*Table A.15.* Pretraining Runtime (hours)

| Name | Training time (hours) |
|---|---|
| *TabPFN* | 7.06 |
| *TabPFN(v2)* | 13.59 |
| *TabPFN(2.5)* | 18.14 |
| *TabICL* | 7.24 |
| *LimiX-2M* | 6.58 |
| *LimiX-16M* | 10.16 |
| *nanoTabPFN* | 12.89 |

## A.7. Synthetic Data Generation (Prior)

For pretraining the models (e.g., *nanoTabPFN* and *nanoTabPFN$_{looped}$*), as well as for pretraining the individual decoders used in the Tabular Logit Lens, we rely on the *TabICL* implementation to generate training priors. The configuration used for prior generation is detailed below:

*Table A.16.* Configuration used for synthetic data generation (prior) with *TabICL*.

| Parameter | Value |
|---|---|
| Batch size per GP | 4 |
| Number of batches | 10,000 |
| Minimum number of features | 2 |
| Maximum number of features | 30 |
| Maximum number of classes | 10 |
| Maximum sequence length | 1024 |
| Log-scaled sequence length | False |
| Sequence length per GP | False |
| Minimum train size ratio | 0.1 |
| Maximum train size ratio | 0.9 |

# B. Experiments and Results

## B.1. Embedding similarity

We additionally report the similarity between embeddings from different layers of each model across benchmarks. As shown in Figure B.1, the patterns are largely consistent between benchmarks.

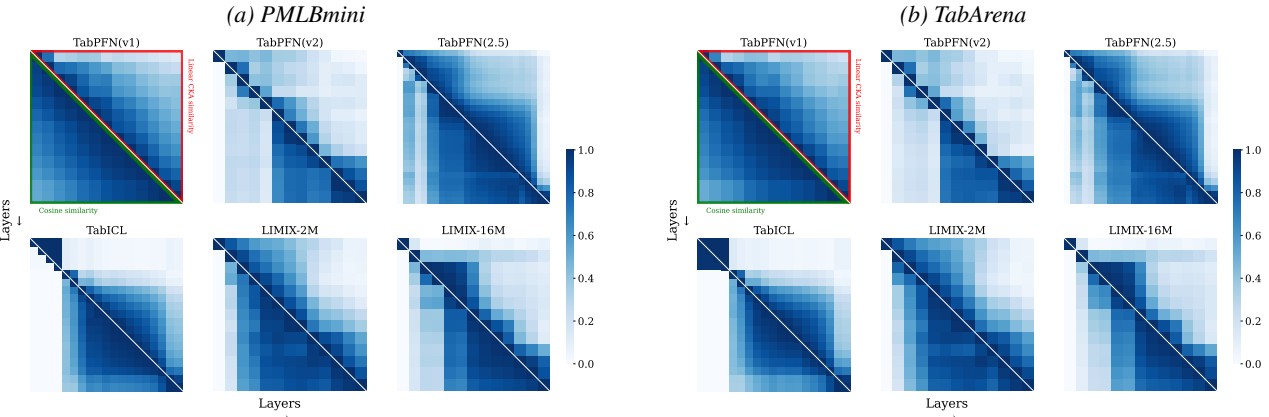

*Figure B.1.* Layer-wise embedding similarity for each model across different benchmarks. Higher values indicate greater similarity between embeddings from the corresponding layers.

## B.2. Separation gap

Here we provide more details about our separation gap.

**Definition.** Let $h_\ell(x_i) \in \mathbb{R}^d$ denote the representation of sample $x_i$ at layer $\ell$, and $y_i$ its class label. We define the sets of within-class and between-class pairs as

$$\mathcal{P}_\ell^{\text{within}} = \{(x_i, x_j) : y_i = y_j, i \neq j\}, \quad \mathcal{P}_\ell^{\text{between}} = \{(x_i, x_j) : y_i \neq y_j\}.$$

Let $d(u, v)$ denote a distance metric between two embeddings. Then, the average within-class and between-class distances at layer $\ell$ are

$$D_\ell^{\text{within}} = \frac{1}{|\mathcal{P}_\ell^{\text{within}}|} \sum_{(x_i, x_j) \in \mathcal{P}_\ell^{\text{within}}} d\big(h_\ell(x_i), h_\ell(x_j)\big), \quad D_\ell^{\text{between}} = \frac{1}{|\mathcal{P}_\ell^{\text{between}}|} \sum_{(x_i, x_j) \in \mathcal{P}_\ell^{\text{between}}} d\big(h_\ell(x_i), h_\ell(x_j)\big).$$

The *separation gap* is then defined as

$$\Delta_\ell = D_\ell^{\text{between}} - D_\ell^{\text{within}}.$$

For example, if $d(u, v)$ is the cosine distance $d_{\cos}(u, v) = 1 - \frac{u \cdot v}{\|u\| \|v\|}$, and for Euclidean distance $d_{\text{E}}(u, v) = \|u - v\|_2$. notably a larger $\Delta_\ell$ indicates stronger separation between classes.

**Separation Gap Across Layers and Benchmarks**. Figures B.2 show the results for cosine distance.

Since the embeddings have very high dimensionality, using distance metrics (especially the Euclidean distance) can be sensitive to noise. To mitigate this, we apply PCA to the embeddings. To fit the PCA, we randomly select 5,000 samples from embeddings across all layers. We then project $h_\ell(x)$ onto the top principal components retaining 95% of the variance, which reduces noise while preserving most of the information in the representation. Figures B.3 and B.4 show the results for different distance metrics after applying PCA. As observed, applying PCA helps reduce noise in the embeddings, leading to a clearer signal and more visible separation between classes.

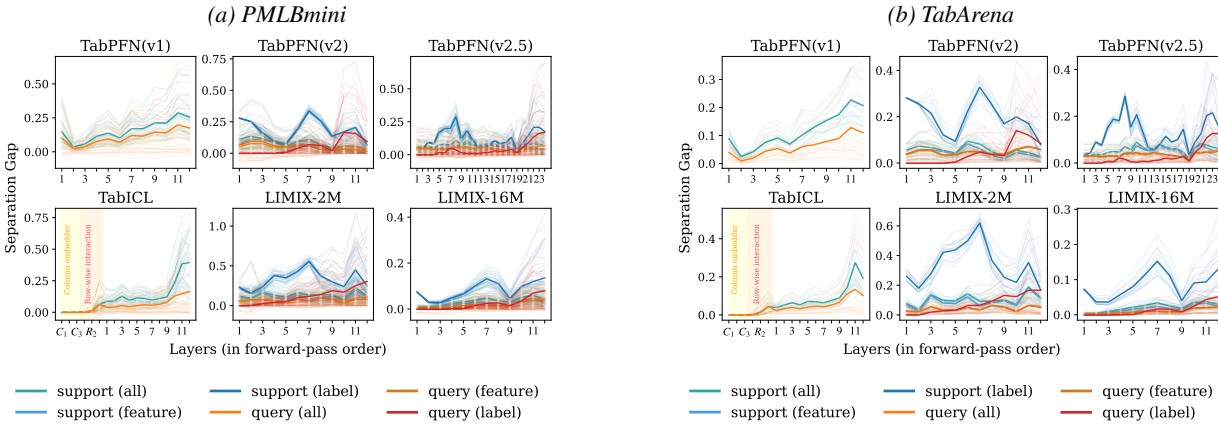

*Figure B.2.* Separation gap between embeddings without PCA using cosine distance. Higher values indicate stronger separation between classes.

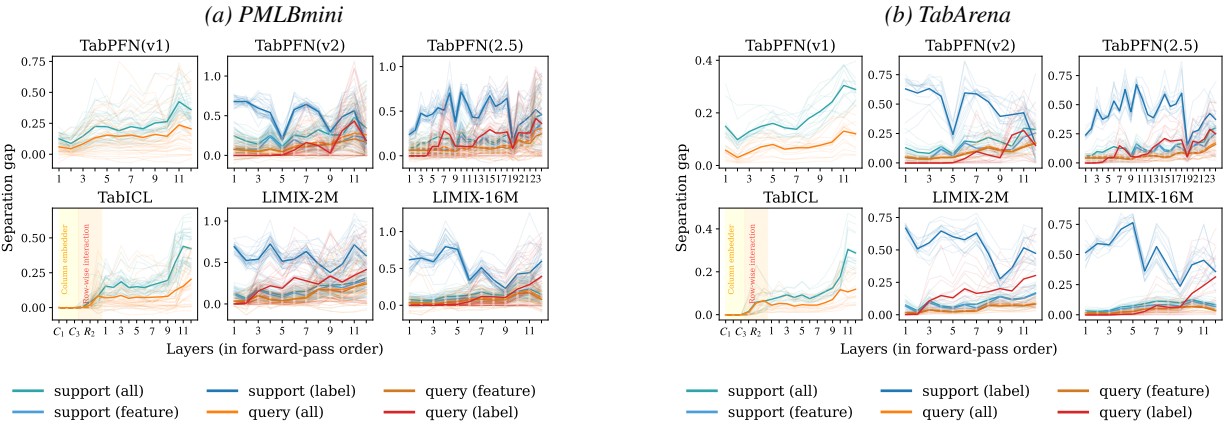

*Figure B.3.* Separation gap after applying PCA, using cosine distance.

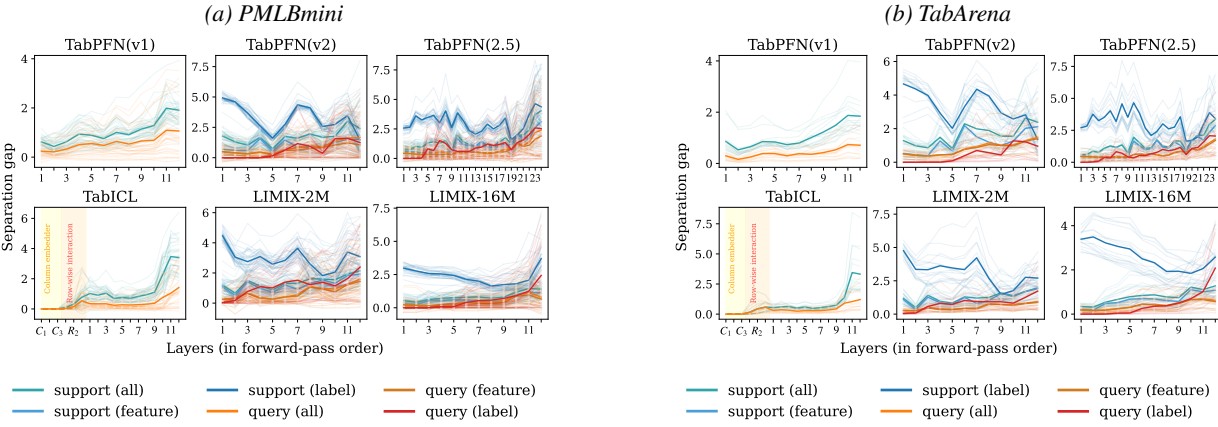

*Figure B.4.* Separation gap after applying PCA, using Euclidean distance.

## B.3. Probing Classifier

As illustrated in Figure B.5, the schematic shows the probing classifier experiment in the ICL setup. Embeddings $h_l$ are extracted from the hidden states at layer $l$ using the training portion of the query set (indicated in blue). A probing classifier, (for example, logistic regression) is trained on these embeddings. For evaluation, embeddings are extracted from the validation portion of the query set at the same layer or other layers (indicated in orange), and the classifier's performance is measured. This setup allows us to assess the extent to which label information from the support set is encoded in each layer's representation.

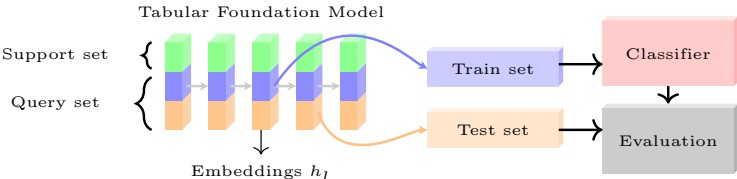

*Figure B.5.* Query set consists of both training and validation samples for probing classifier. Blue corresponds to the training set, and orange corresponds to the validation set.

We evaluate multiple probing classifiers, including logistic regression, k-Nearest Neighbors (KNN), and Linear Discriminant Analysis (LDA), as well as a fine-tuned original decoder. For each classifier, we report the normalized ROC-AUC on the *PMLBmini* and *TabArena* benchmarks.

Figures B.7, B.6 and B.8 show the results for KNN, logistic regression, and LDA probing classifiers, respectively, trained on embeddings extracted from different layers of the models. Figure B.9 presents the results for the fine-tuned original decoder used as a probing classifier.

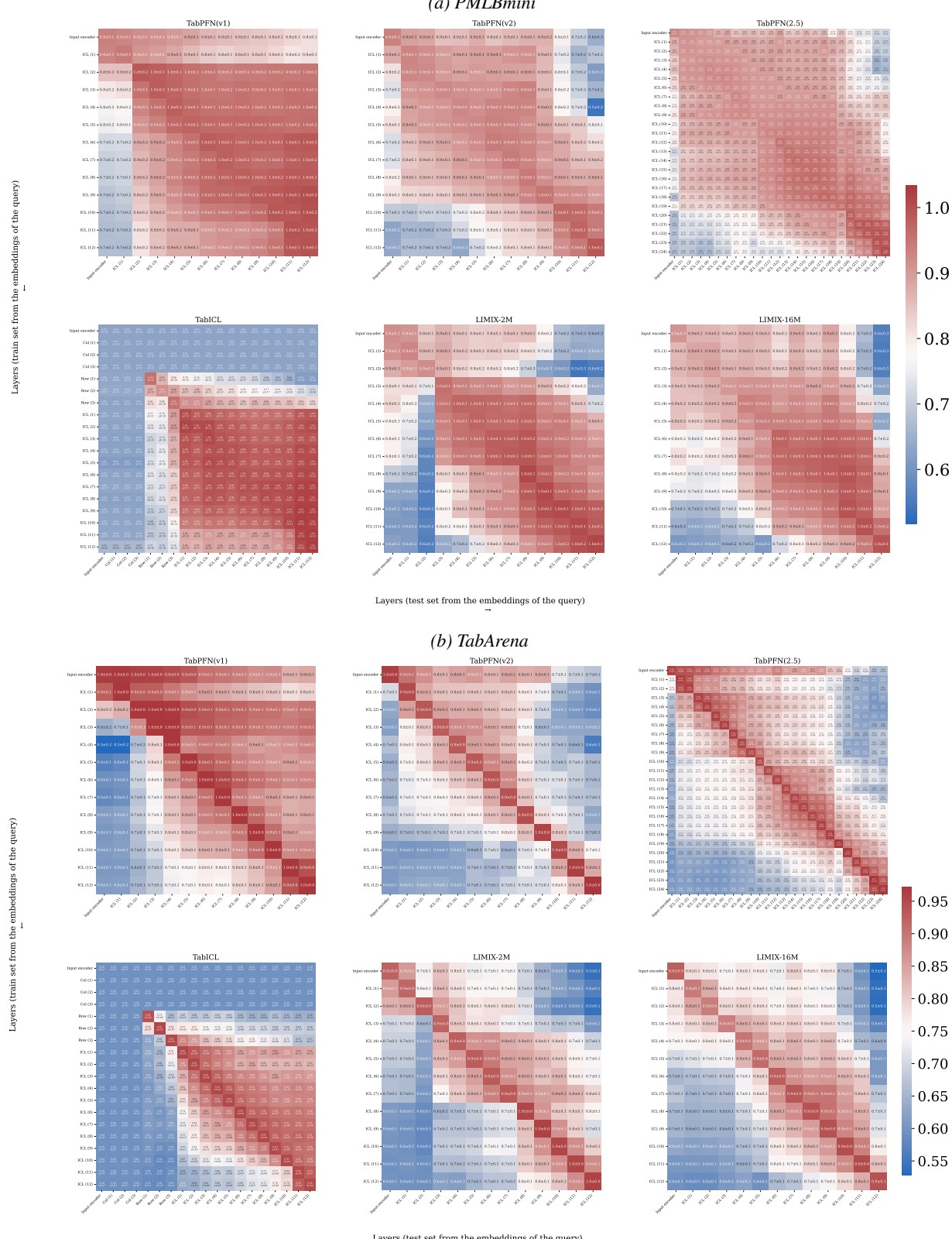

*Figure B.6.* Normalized AUC for logistic regression as a probing classifier trained on embeddings at different layers of the models for each benchmark.

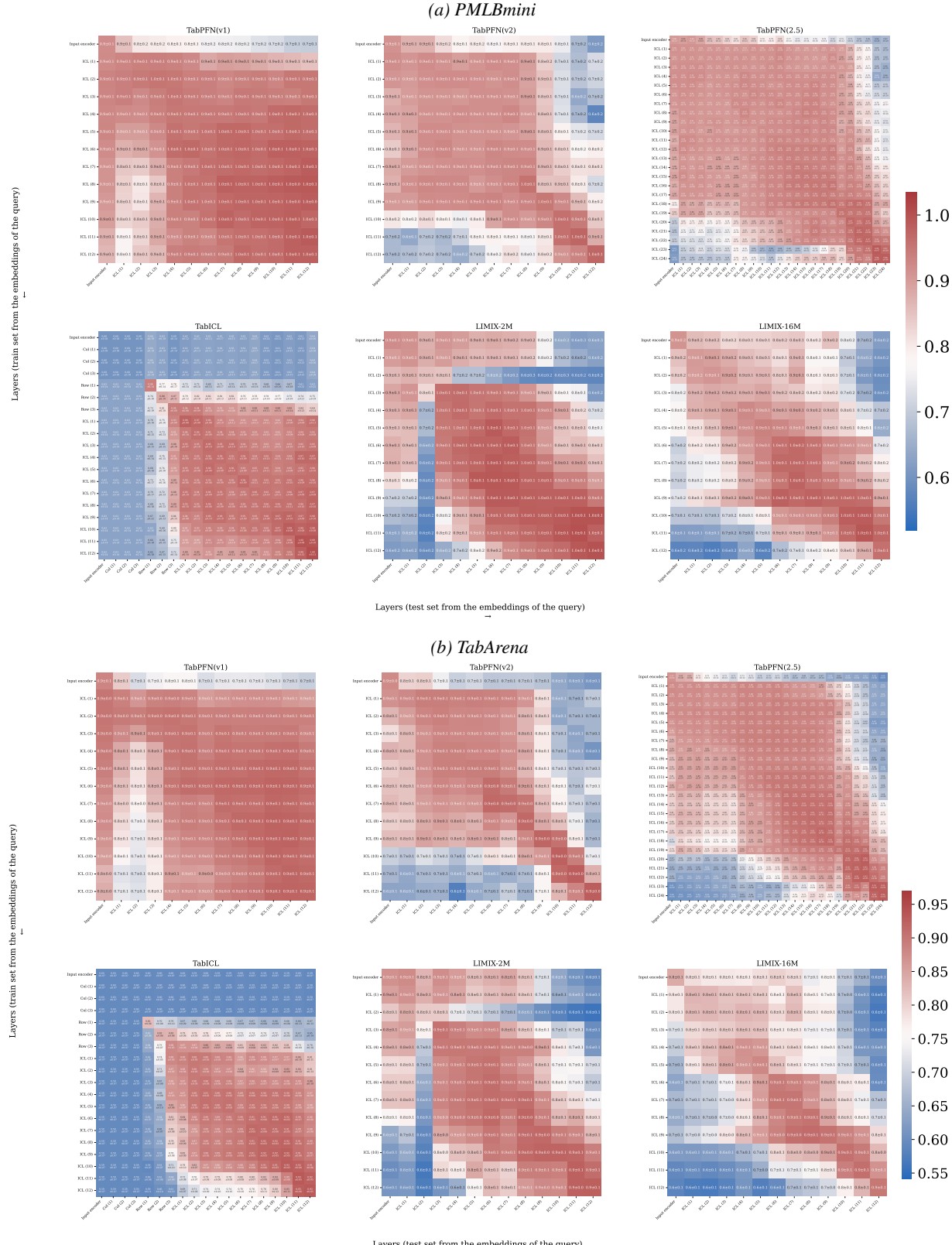

*Figure B.7.* Normalized AUC for k-Nearest Neighbors (KNN) as a probing classifier trained on embeddings at different layers of the models.

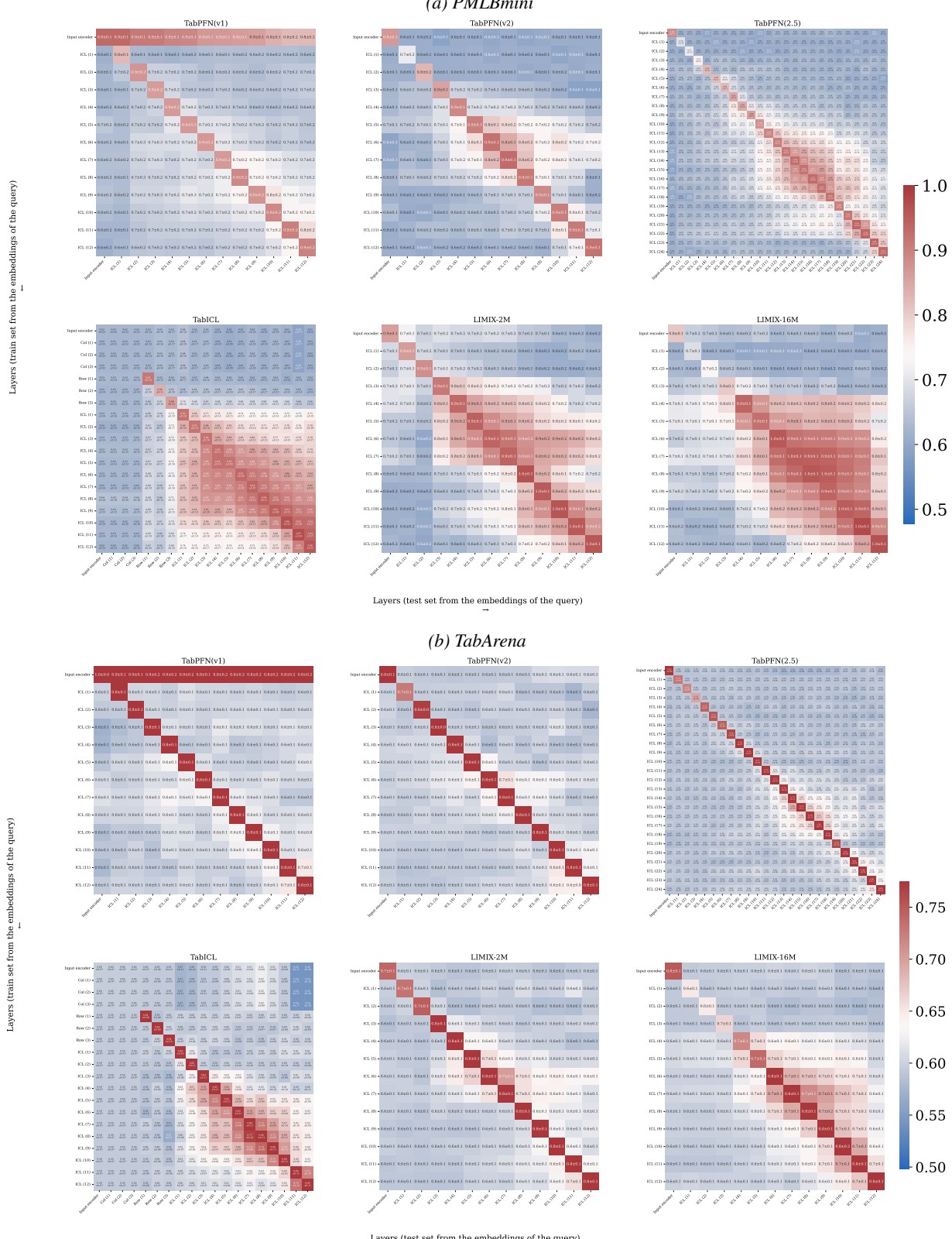

*Figure B.8.* Normalized AUC for Linear Discriminant Analysis (LDA) as a probing classifier trained on embeddings at different layers of the models.

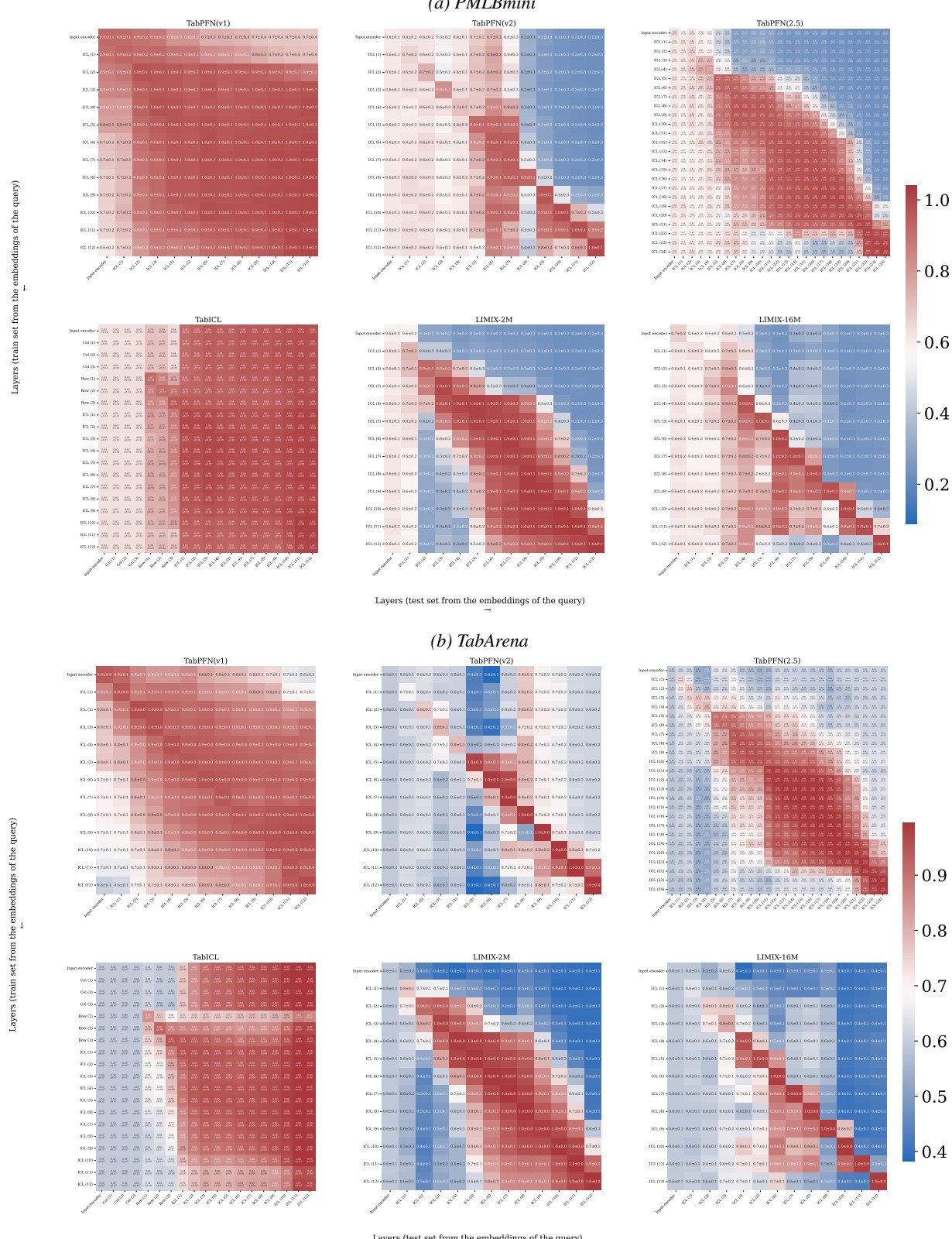

*Figure B.9.* Normalized AUC for fine-tuned original decoder as a probing classifier trained on embeddings at different layers of the models.

## B.4. Tabular Logit Lens

Here, we present the results of the early-exit strategy. In this approach, after each layer, the embeddings are passed to the decoder to produce a prediction. Figure B.10 shows the performance of each model at different early-exit points, while Figure B.11 reports the corresponding prediction entropy. Higher entropy indicates that the model is less confident, with output probabilities being more evenly distributed across classes. Furthermore, in Figure B.12, we report ROC-AUC and balanced accuracy across layers. From the middle to the final layers, although the ROC-AUC of the original decoder matches that of the corresponding individual decoders, its balanced accuracy remains substantially lower. This discrepancy indicates that the original decoder's predictions are poorly calibrated: while ranking performance (ROC-AUC) is preserved, decision-threshold–dependent metrics such as balanced accuracy degrade. This need for calibration is consistent with our observations based on prediction entropy in Figure B.11.

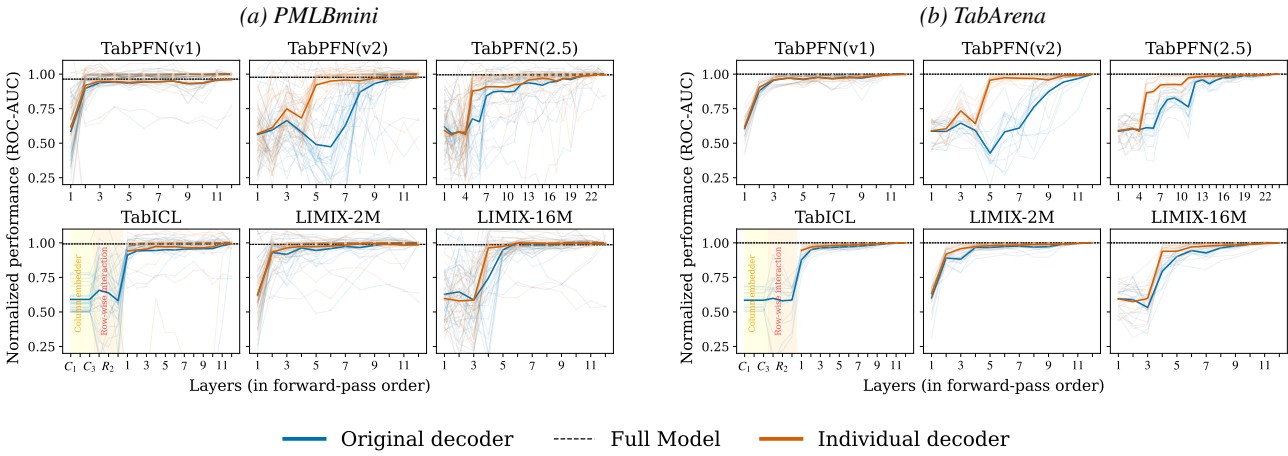

*Figure B.10.* Effect of the tabular logit lens experiment on the tabular foundation model's performance.

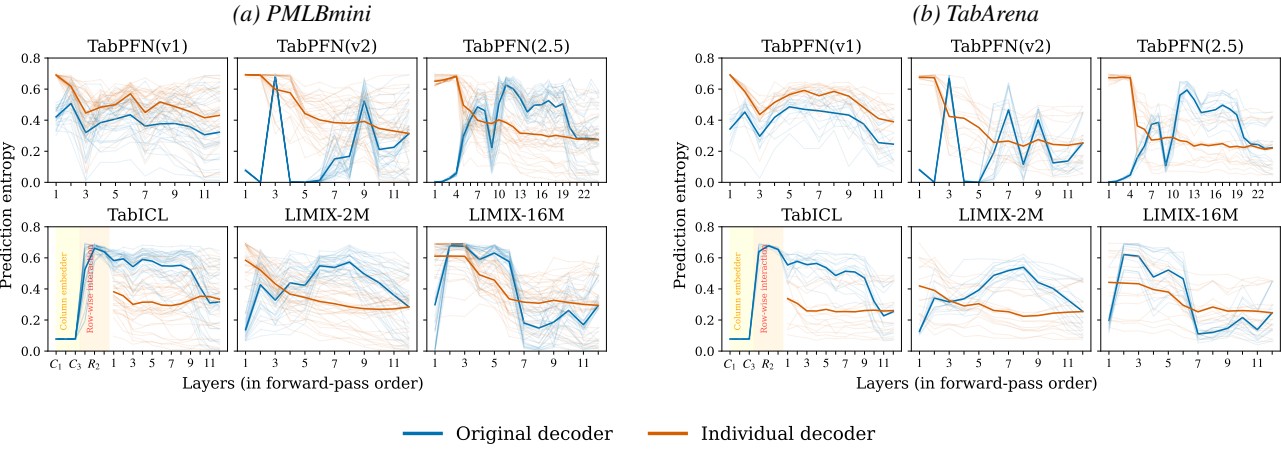

*Figure B.11.* Effect of the tabular logit lens experiment on prediction entropy of the tabular foundation model.

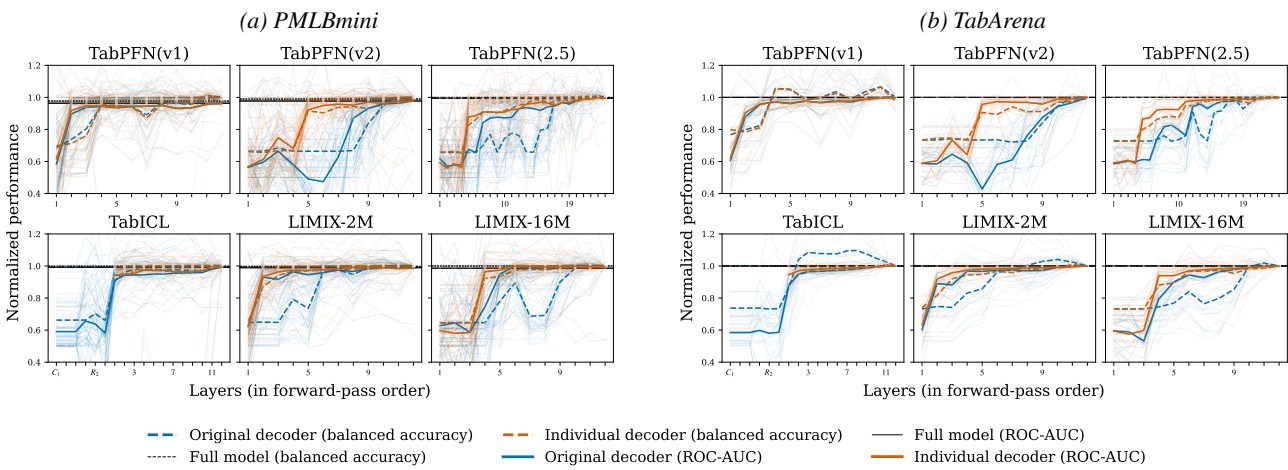

*Figure B.12.* Comparison of ROC-AUC and balanced accuracy across layers on *PMLBmini* and *TabArena*, illustrating prediction calibration through tabular logit lens.

## B.5. Layer ablation

We do layer ablations as illustrated in Figure 2 and report detailed results of the layer ablation experiment. As shown in Figure B.13, the effects of layer ablation vary slightly across benchmarks. In particular, *TabArena* exhibits greater sensitivity to layer ablation compared to *PMLBmini*. Figure B.14 shows the corresponding prediction entropy, providing insight into the model's confidence at each layer. As observed, for *TabPFN(2.5)* the final layers are primarily responsible for the model's predictive certainty, whereas this effect is less pronounced in the other models. Furthermore, to facilitate a clearer comparison between benchmarks, we provide Figure B.15.

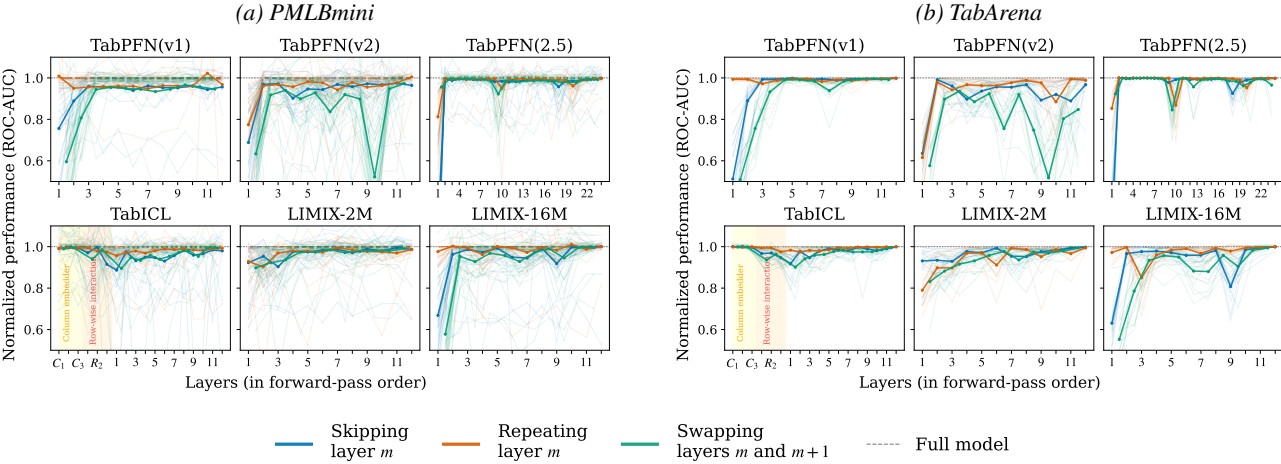

*Figure B.13.* Effect of the layer ablation on prediction on the tabular foundation model's performance.

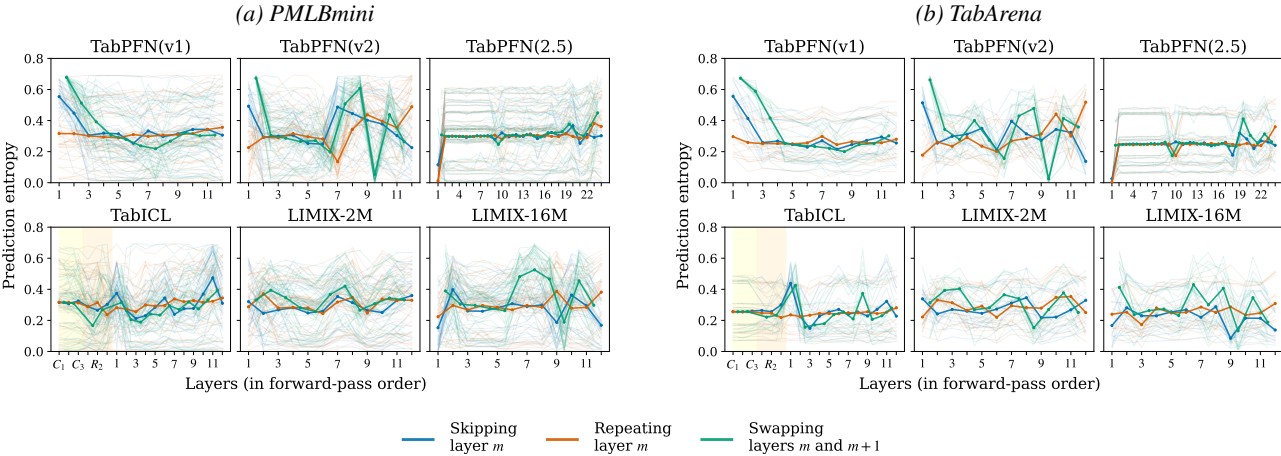

*Figure B.14.* Effect of the layer ablation on prediction entropy for the tabular foundation model's performance.

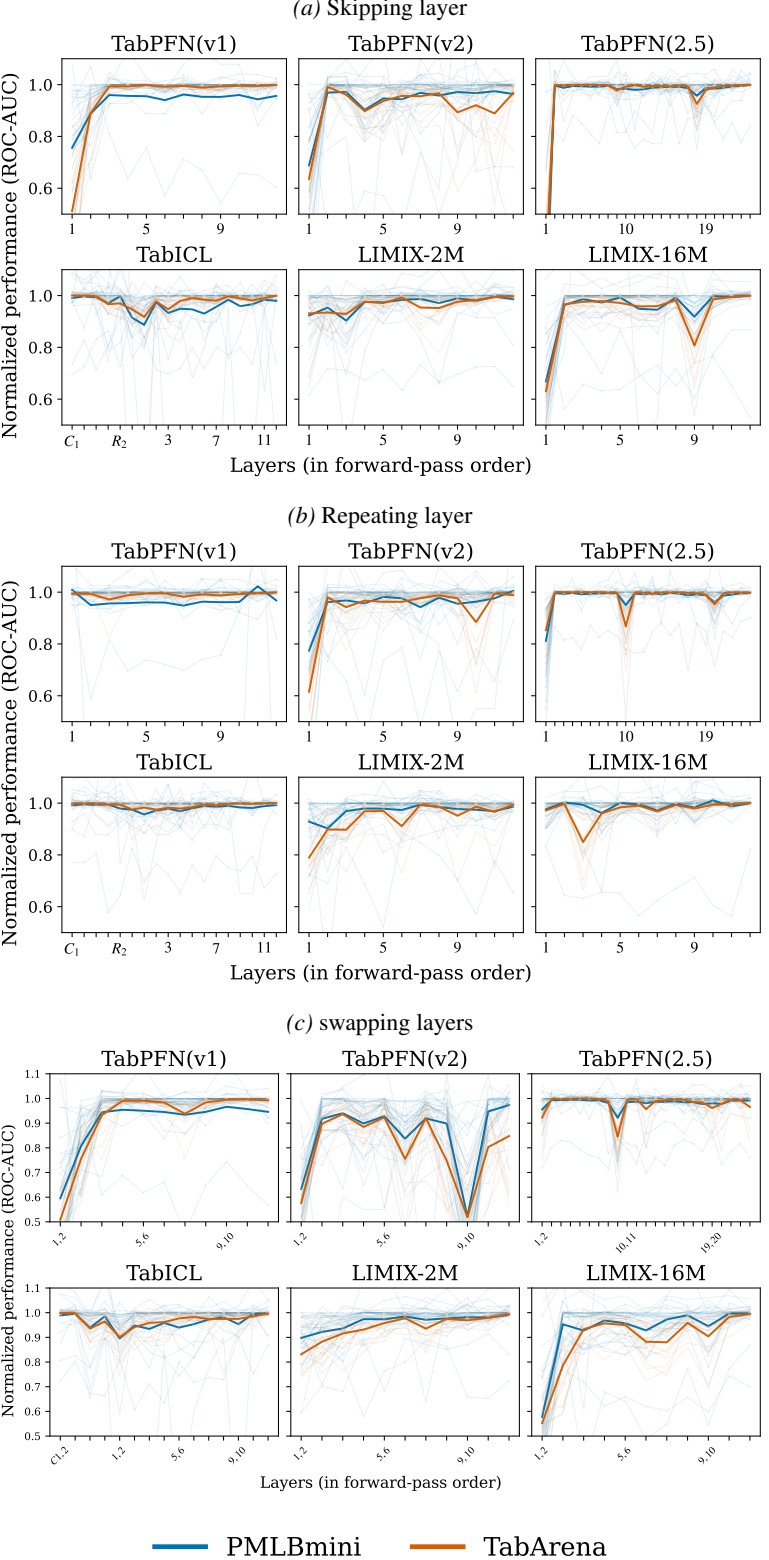

*Figure B.15.* Effect of the layer ablation across benchmarks.

## B.6. Self-repair

We provide more detailed results illustrating the self-repair phenomenon. Figures B.16, B.17 and B.18 show layer-wise performance with and without layer ablation. For the top plots, the solid black line represents performance without any intervention. After ablating a layer, the change in performance is indicated by dashed lines, and it continues with the colored lines, from blue (early) to orange (late), that show intermediate performance following the layer ablations. This visualization highlights how performance drops after an intervention and recovers in subsequent layers, indicating self-repair. In the bottom plots, we compare the change in performance between the final prediction and the immediate prediction after a layer ablation. Each dot corresponds to a dataset in the benchmark, and colors indicate the layer index, from blue (early) to orange (late). We observe instances where the final performance remains largely unchanged despite a significant drop in immediate performance, indicating self-repair (e.g., the green points in *TabPFN(2.5)*). Conversely, in *TabPFN(2.5)*, early-layer interventions (blue) typically do not recover, suggesting that self-repair does not occur in these cases.

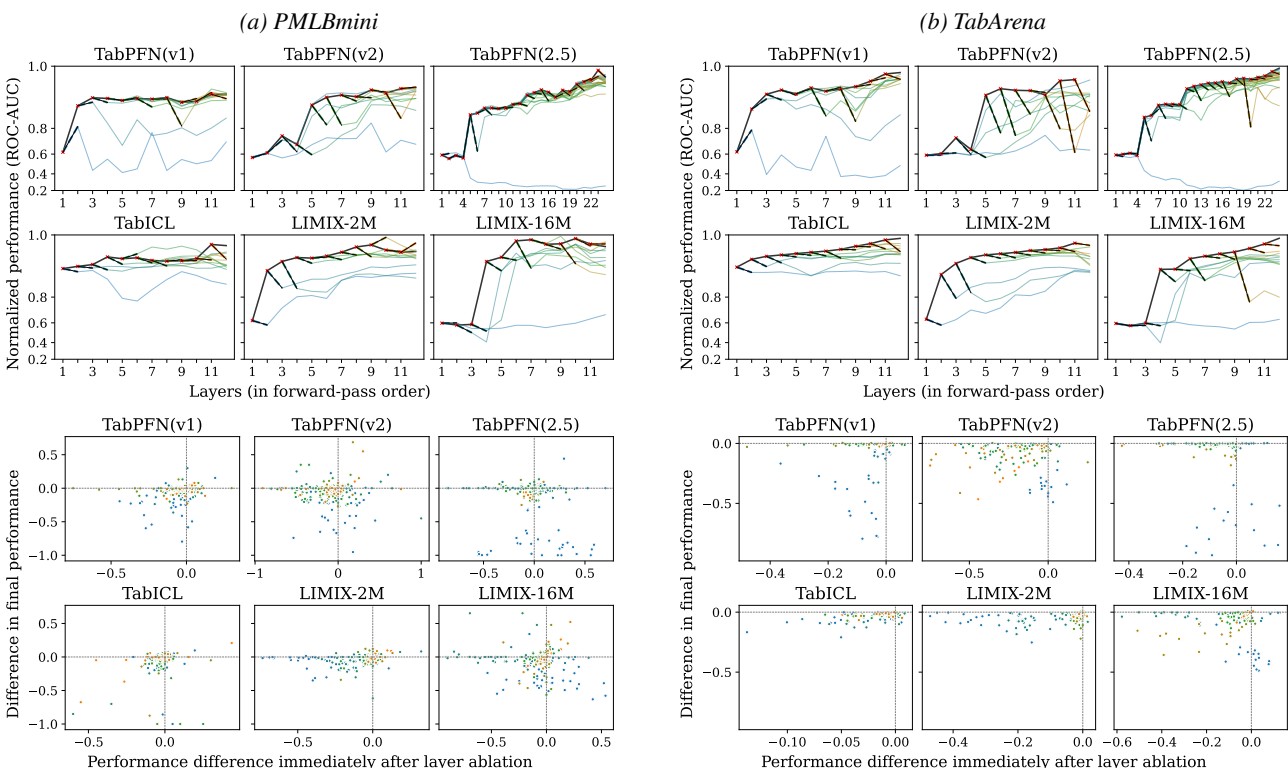

*Figure B.16.* Layer-wise self-repair following a skipped layer.

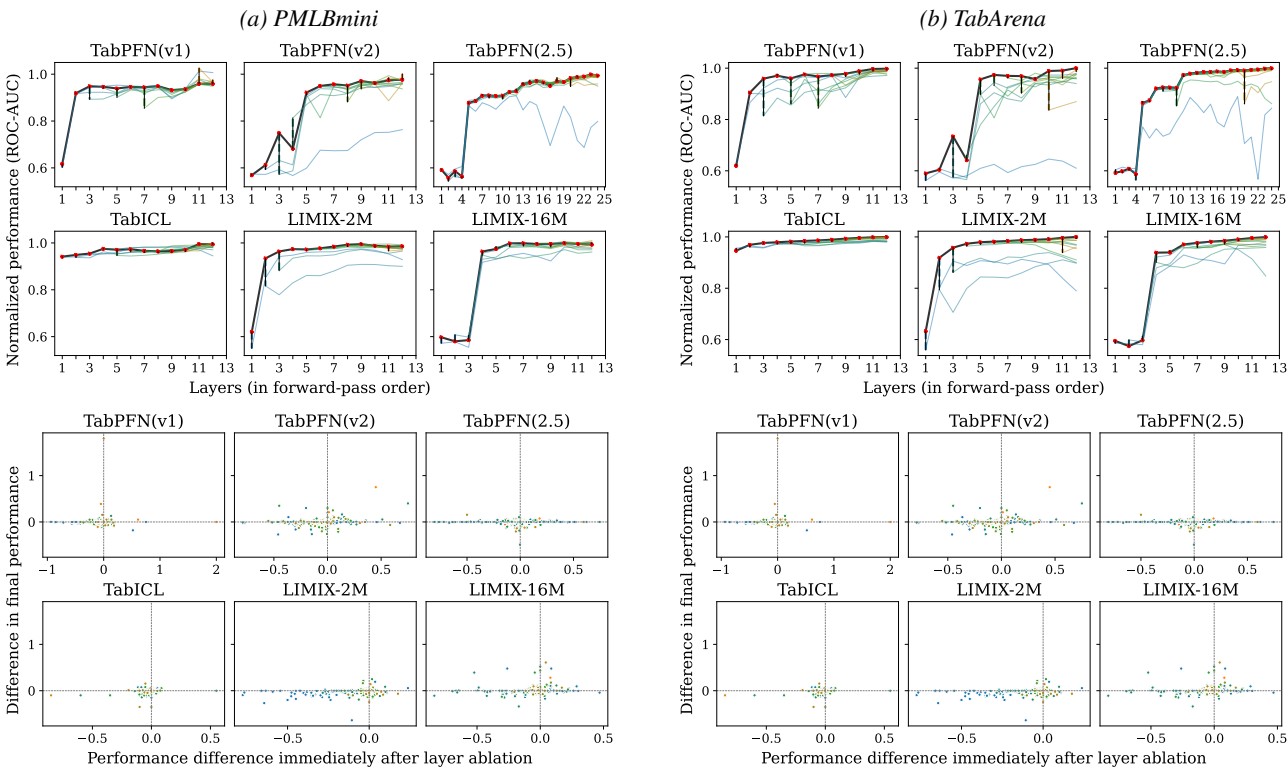

*Figure B.17.* Layer-wise self-repair following a repeated layer.

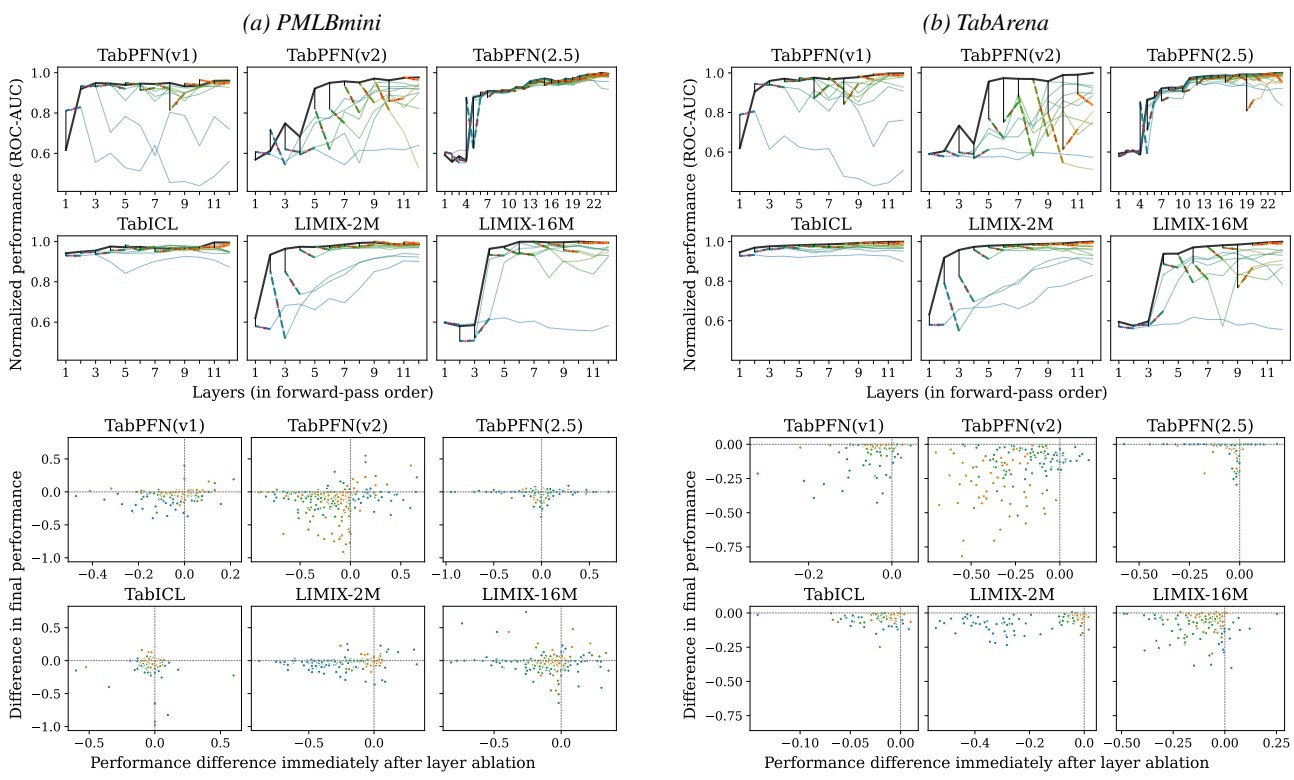

*Figure B.18.* Layer-wise self-repair following swapped layers.

# C. Further Discussion

In this section, we provide a conceptual interpretation of the internal inference process of TFMs by drawing an analogy with the well-studied inference stages of Large Language Models (LLMs). Figure C.1 and Table C.1 summarize this comparison by aligning each inference stage with the supporting experimental evidence, highlighting both shared structural principles and key differences induced by the tabular prediction setting.

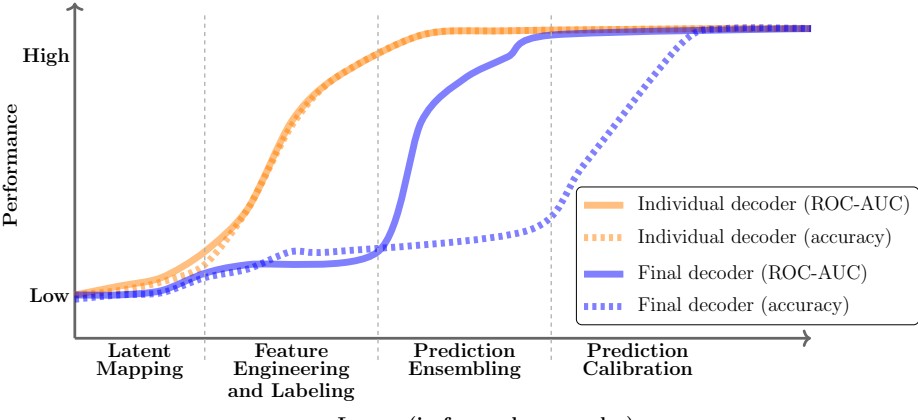

*Figure C.1.* Internal inference process of TFMs.

*Table C.1.* Comparison of inference stages: LLMs versus TFMs

| | LLMs | | TFMs | | |
|---|---|---|---|---|---|
| Stage | Stage | Function | Stage | Function | Evidences |
| 1 | Detokenization | Use local contextual information to map raw tokens into coherent representations | Latent Mapping | Extension of the input encoder; maps input embeddings into coherent representations | 1) Low performance on tabular logit lens (Figure 6) and high sensitivity to layer ablations (Figure 7). 2) Robustness of skipping early layers in models with better input encoders (Figure 7). |
| 2 | Feature Engineering | Iteratively build feature representations depending on token context | Feature Engineering and Labeling | Iteratively build feature representations depending on labels | 1) Enhancing the distinction between intra-class and inter-class feature distributions (Figure 4). 2) Sharp rise in early-exit accuracy espcially for individual decoders (Figure 6), due to internal label embeddings and their iterative adjustment (see red line in Figure 4). |
| 3 | Prediction Ensembling | Convert semantic features into plausible next-token predictions via an iterative ensemble | Prediction Ensembling | Ensemble and transform internal label embeddings into decoder-ready predictions. | In early-exit, individual decoder performance saturates while the original decoder continues to improve, indicating feature alignment with the decoder (Figure 6). |
| 4 | Residual Calibration | Eliminate obsolete features and form the final output distribution | Prediction Calibration | Calibrate the prediction for the original decoder | 1) Prediction entropy continues to decrease (or change) despite stable performance (Figure. B.11.), particularly for the original decoder, even after both individual and original decoders have reached high predictive performance. 2) For the original decoder, balanced accuracy exhibits a delayed improvement, with a noticeable jump occurring only after ROC-AUC has already reached a strong performance level (Figure B.12) |

# D. Multiclass Experiments

In this section, we extend our analysis to multiclass classification tasks on the *TabArena* benchmark. Our main experiments focus on binary classification; here, we verify whether the observed inference dynamics generalize to settings with more than two classes. We report balanced accuracy and multiclass ROC-AUC as the metrics.

## D.1. Datasets

For the multiclass experiments, we use all multiclass classification tasks from *TabArena* that satisfy the model constraints ($\leq$ 10,000 samples and $\leq$ 100 features), resulting in 6 datasets. We apply the same cross-validation protocol as in the binary setting (see Appendix A.5).

## D.2. Embedding Similarity

Figure D.1 shows the layer-wise embedding similarity for each model on multiclass *TabArena* tasks. The block structure observed in the binary setting (Section 3) is similarly visible here.

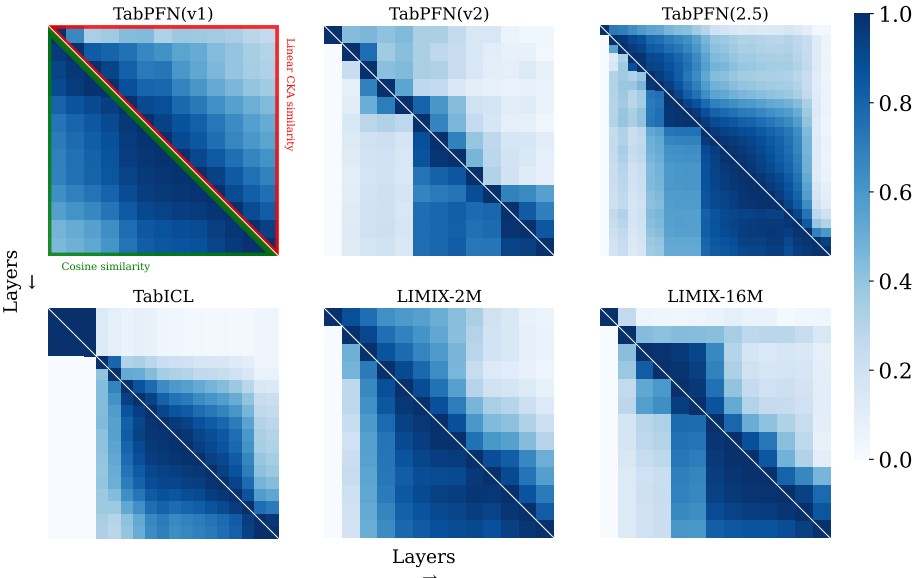

*Figure D.1.* Layer-wise embedding similarity (upper triangular: linear CKA; lower triangular: cosine similarity) for multiclass *TabArena* tasks.

## D.3. Separation Gap

Figures D.2 and D.3 show the separation gap for the multiclass setting using cosine and Euclidean distances, respectively. Consistent with the binary case, the separation gap increases with depth across all models, providing further evidence for the iterative inference hypothesis. The general trend of models incrementally increasing the inter-class distance relative to the intra-class distance holds in the multiclass regime.

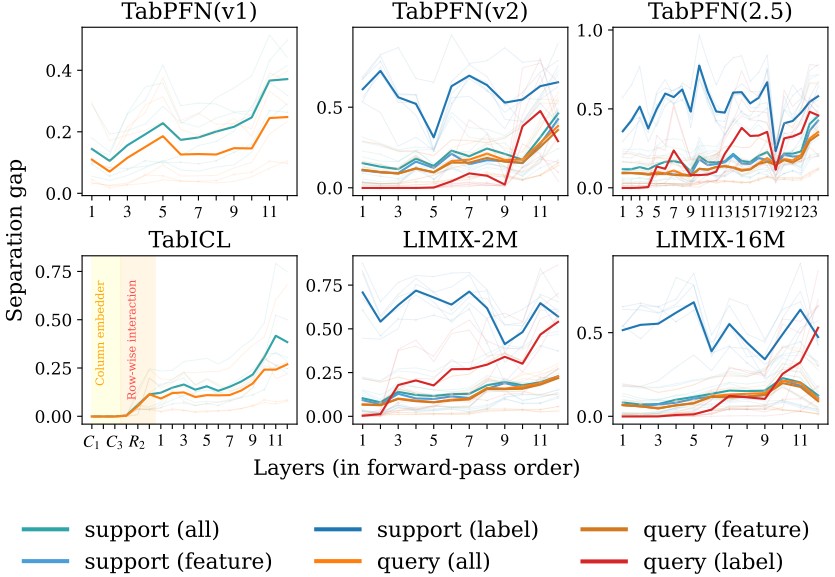

*Figure D.2.* Separation gap (cosine distance, after PCA) across layers for multiclass *TabArena* tasks. Bold lines indicate the average across tasks; thin lines represent individual datasets.

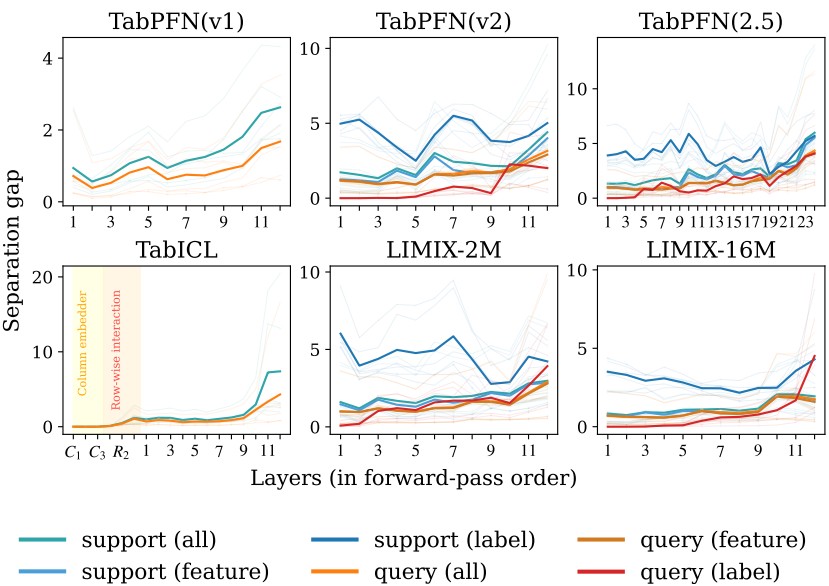

*Figure D.3.* Separation gap (Euclidean distance, after PCA) across layers for multiclass *TabArena* tasks.

## D.4. Probing Classifiers

Figures D.4 report the normalized balanced accuracy of probing classifiers trained on embeddings at different layers of each model, for multiclass *TabArena* tasks. The asymmetric transfer pattern identified in the binary setting (Takeaway of Experiment ③) persists: a classifier trained on layer $i$ generally transfers better to later layers $j > i$ than the reverse, confirming that each layer cumulatively enriches the representation.

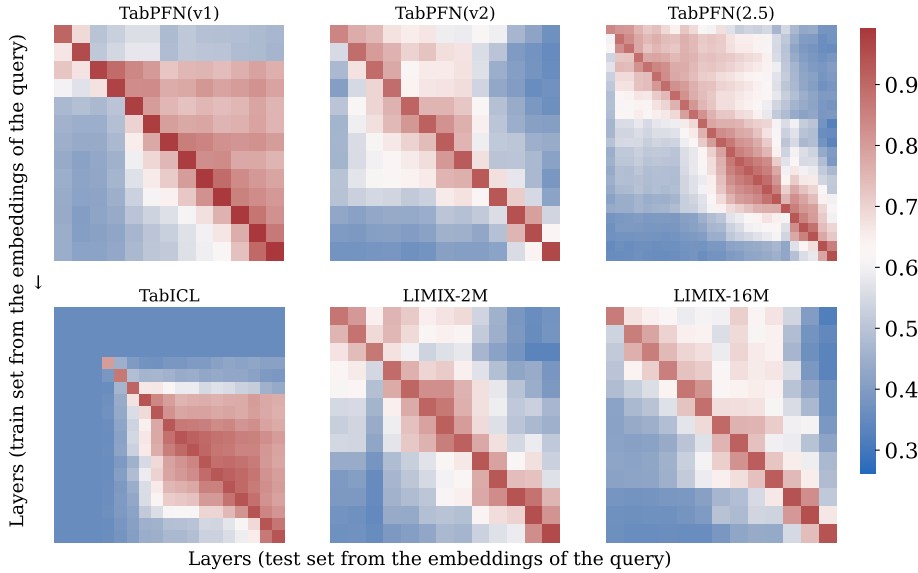

*Figure D.4.* Normalized balanced accuracy for logistic regression as a probing classifier, multiclass *TabArena*.

## D.5. Tabular Logit Lens

Figure D.5 reports the performance of individual decoders at each layer for multiclass *TabArena* tasks and compares ROC-AUC and balanced accuracy across layers. Figure D.6 shows the corresponding prediction entropy.

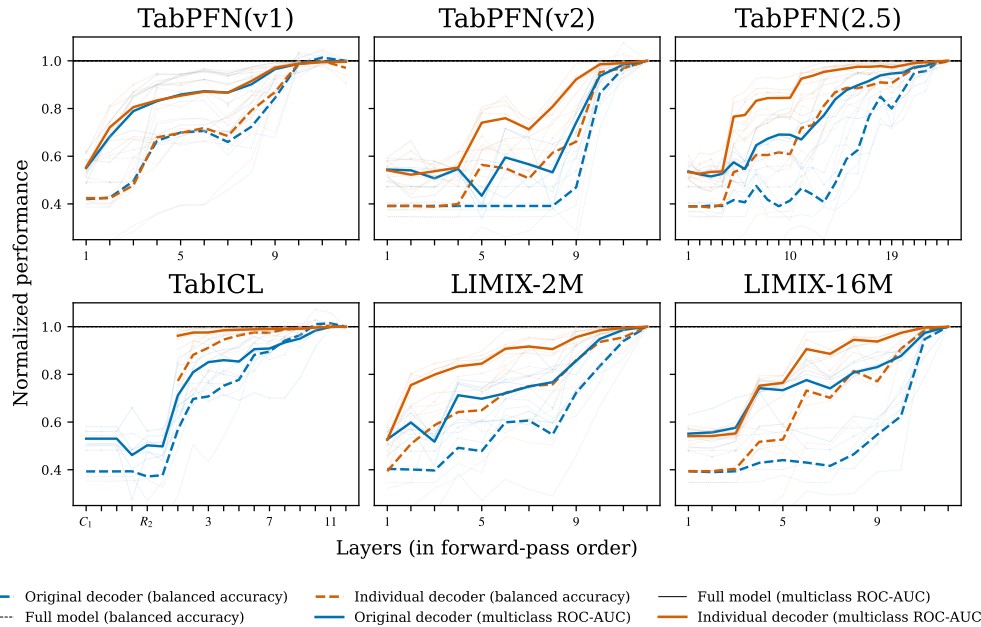

*Figure D.5.* Comparison of ROC-AUC and balanced accuracy across layers for multiclass *TabArena* tasks, illustrating prediction calibration via the tabular logit lens.

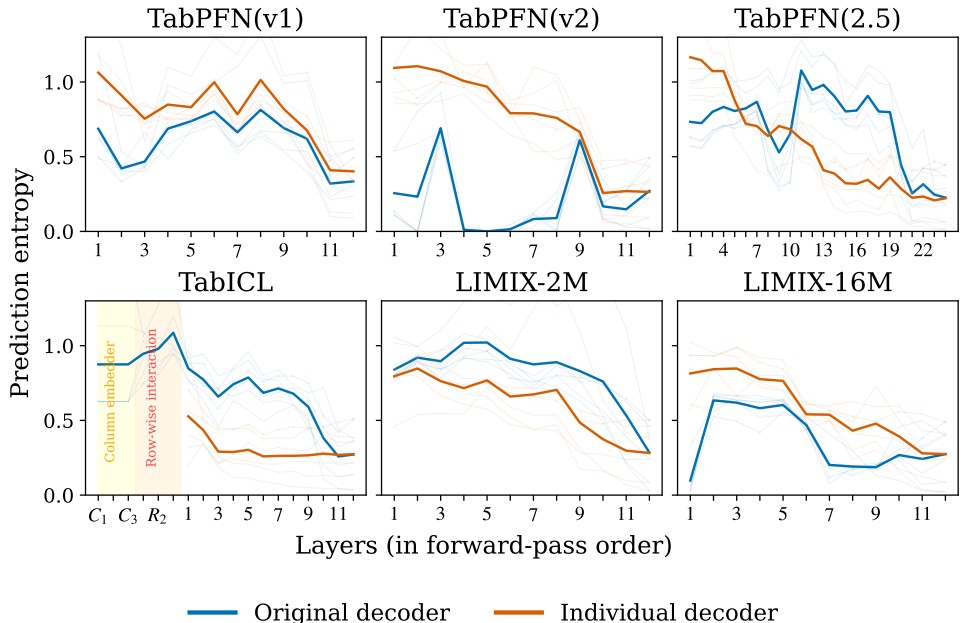

*Figure D.6.* Prediction entropy across layers for multiclass *TabArena* tasks.

## D.6. Layer Ablation

Figures D.7 and D.8 show the effect of layer ablations and the corresponding prediction entropy for multiclass *TabArena* tasks. The sensitivity profile is consistent with the binary setting: early layers are the most sensitive to ablation, middle and later layers exhibit robustness, and repeating certain layers provides marginal improvement. The overall pattern confirms that the observed dynamics are not an artifact of the binary classification objective.

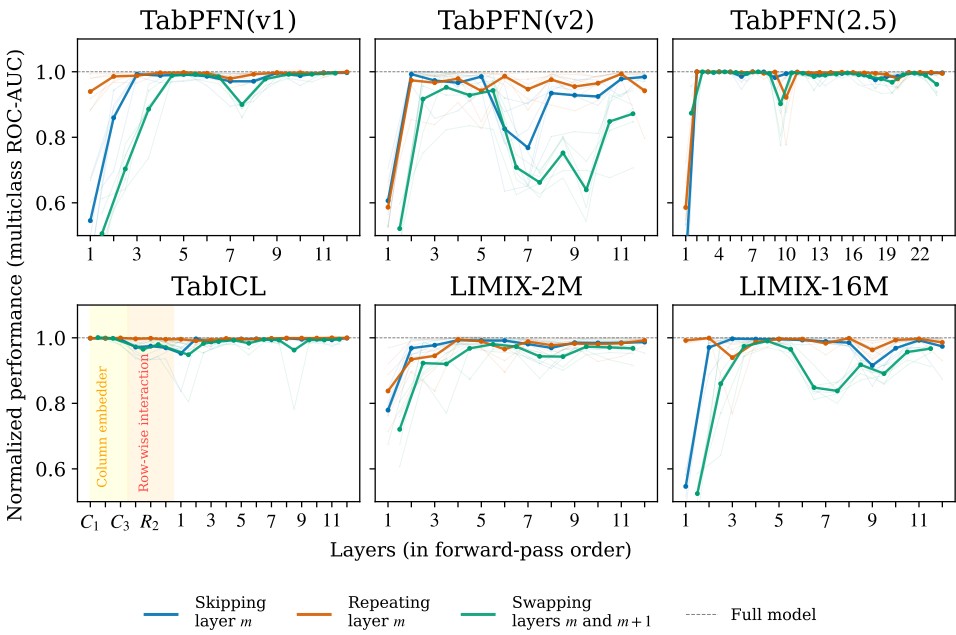

*Figure D.7.* Effect of layer ablation on model performance for multiclass *TabArena* tasks.

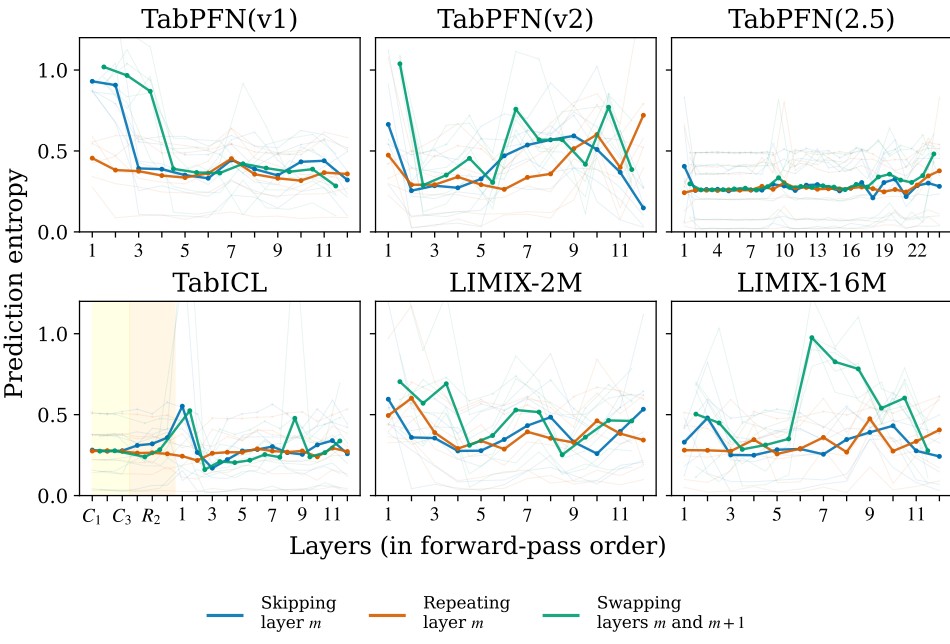

*Figure D.8.* Prediction entropy under layer ablation for multiclass *TabArena* tasks.

## D.7. Self-Repair

Figure D.9 shows the self-repair analysis for the layer-skipping ablation in the multiclass setting. As in the binary case, early-layer ablations are not recovered in subsequent layers, whereas middle and later layers exhibit clear self-repair. Figures D.10 and D.11 present the analogous results for layer repetition and swapping, respectively.

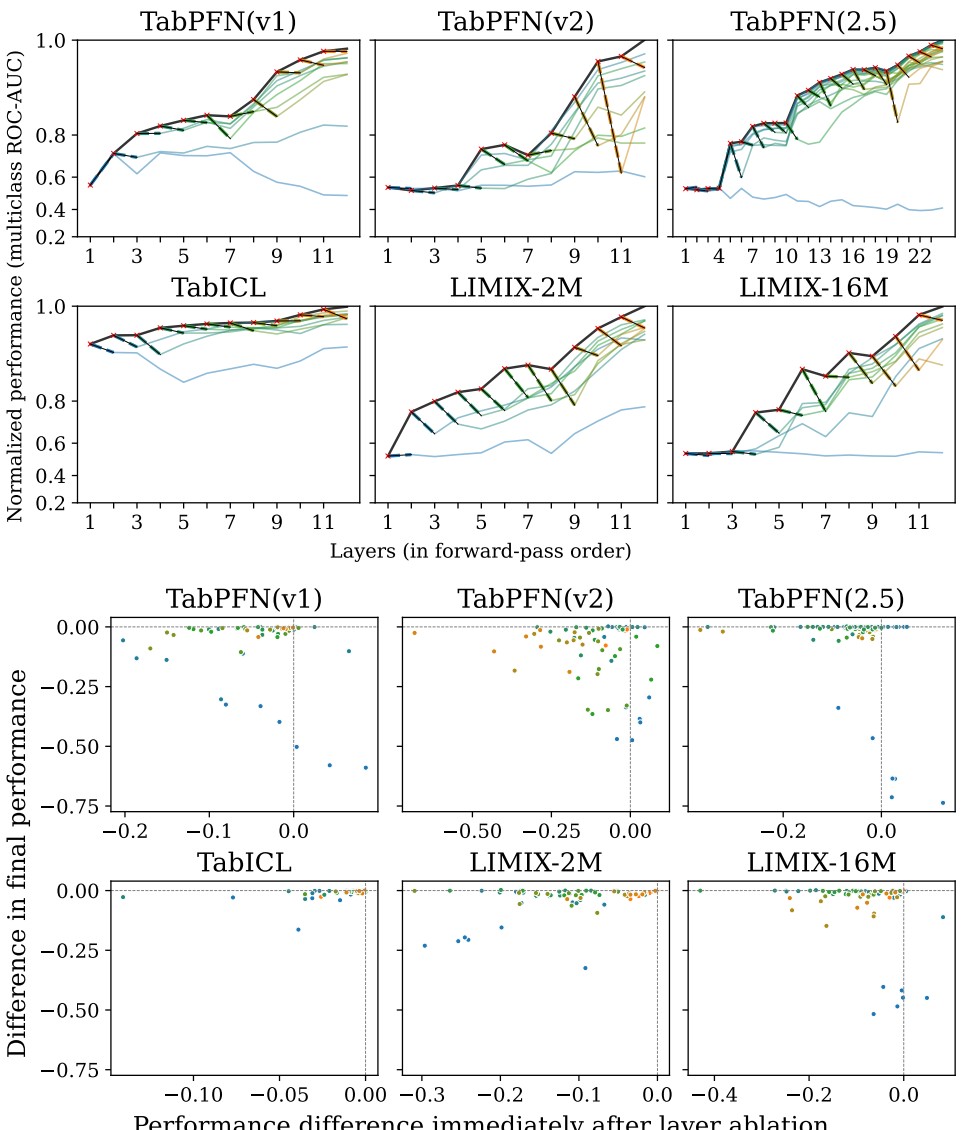

*Figure D.9.* Layer-wise self-repair following a skipped layer, multiclass *TabArena*.

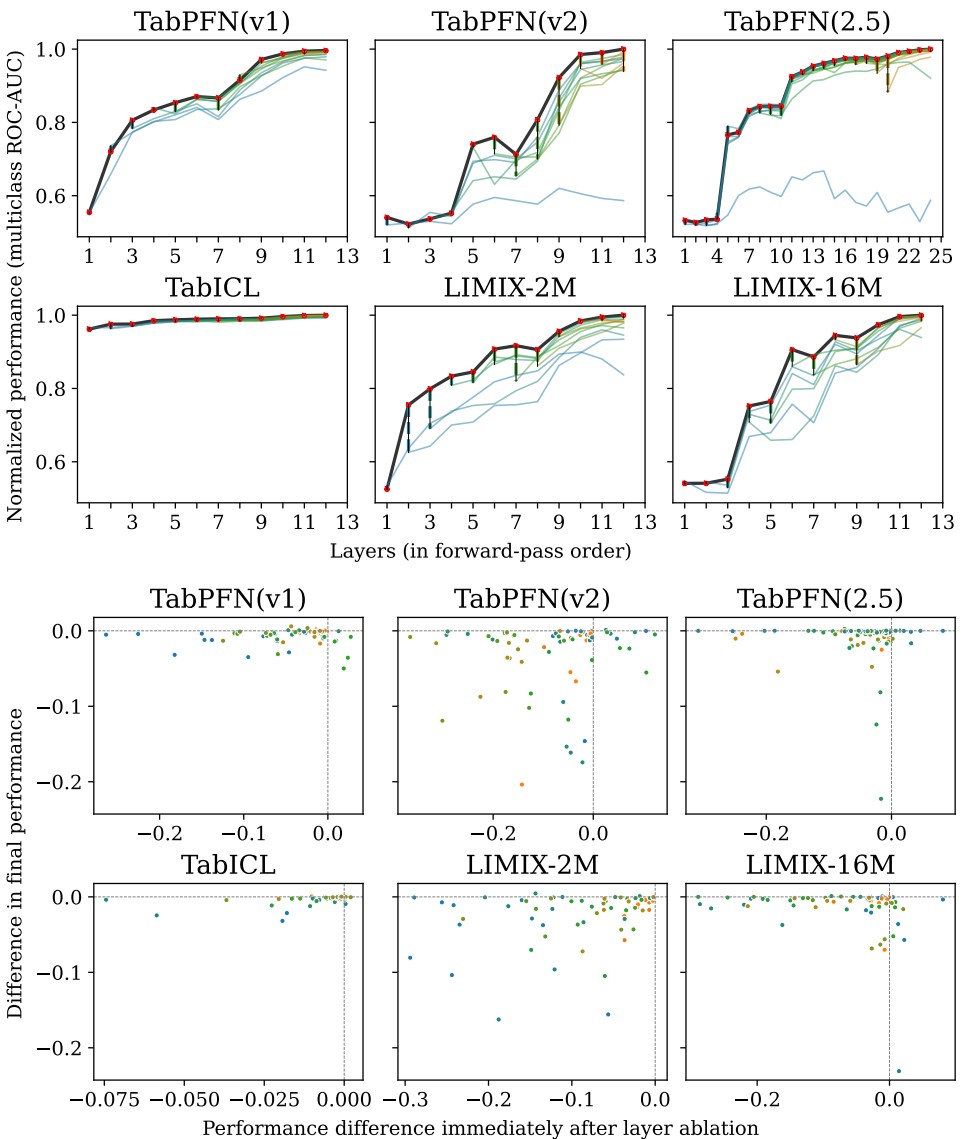

*Figure D.10.* Layer-wise self-repair following a repeated layer, multiclass *TabArena*.

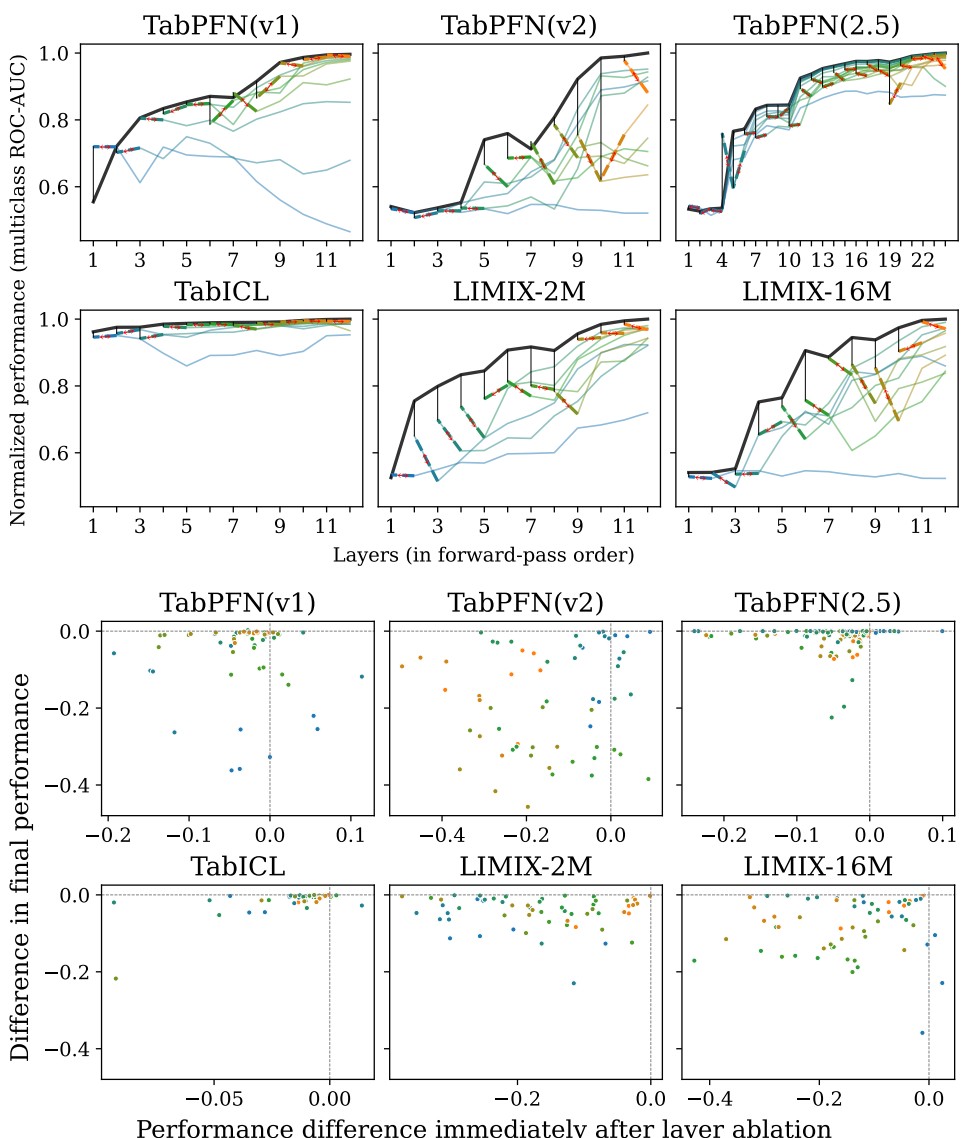

*Figure D.11.* Layer-wise self-repair following swapped layers, multiclass *TabArena*.

## D.8. Is One Layer Enough?

Figure D.12 extends the looped transformer evaluation to the multiclass *TabArena* setting. The *nanoTabPFN$_{looped}$* model achieves performance comparable to the six-layer *nanoTabPFN* baseline, consistent with the binary results. This supports the conclusion that iterative reuse of a single transformer block is a viable strategy for parameter-efficient TFM design beyond binary classification. We also add 20 multiclass tasks from the OpenML CC18 suite that satisfy the model constraints ($\leq 10,000$ samples and $\leq 100$ features).

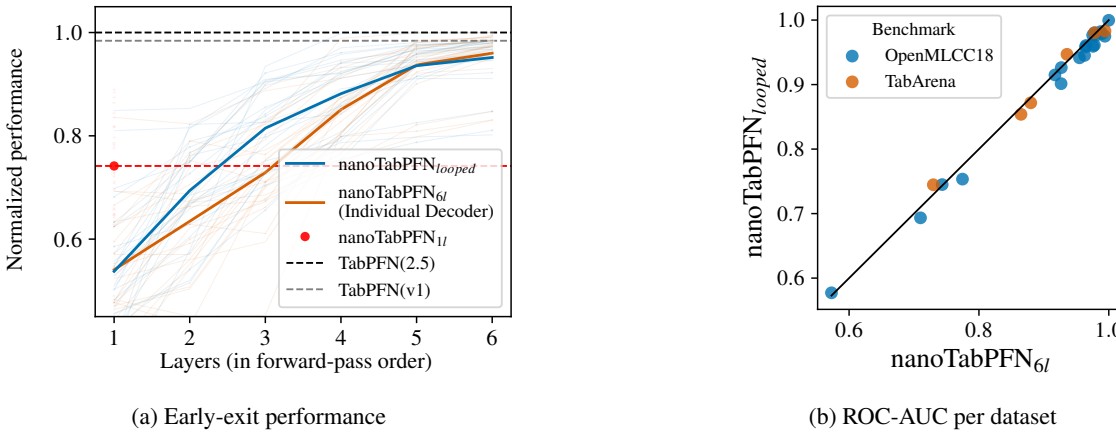

(a) Early-exit performance        (b) ROC-AUC per dataset

*Figure D.12.* Multiclass *TabArena* evaluation of *nanoTabPFN$_{6l}$*, *nanoTabPFN$_{1l}$*, and *nanoTabPFN$_{looped}$*.

# E. Regression Experiments

In this section, we extend our analysis to regression tasks on the *TabArena* benchmark to verify that our main findings are not specific to the classification setting. We adapted the experiments as follows: we include all TFMs that support regression, namely *LimiX-2M*, *LimiX-16M*, *TabPFN(v2)*, and *TabPFN(2.5)*; we use negative RMSE as the primary performance metric and additionally report Spearman's rank correlation, which is invariant to calibration; the separation gap experiment is extended to regression by treating it as a classification problem (discretizing regression values into $K = 10$ bins); we adapt *TabICL* priors to regression tasks for pretraining individual decoders; and we include all six regression tasks in *TabArena* with fewer than 100 features and 10,000 samples, using the repetitions and folds specified by *TabArena*. Overall, the results on regression tasks are consistent with and support our main findings.

## E.1. Embedding Similarity

Figure E.1 shows the layer-wise embedding similarity on regression tasks. As in the classification setting, TFMs form blocks in which the embeddings remain similar across adjacent layers.

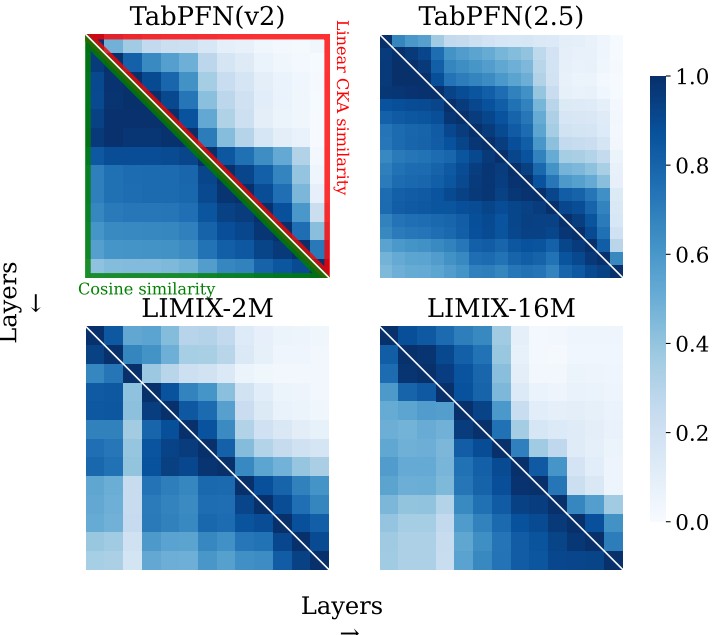

*Figure E.1.* Layer-wise embedding similarity (upper triangular: linear CKA; lower triangular: cosine similarity) for regression *TabArena* tasks.

## E.2. Separation Gap

We extend the separation gap metric to regression by discretizing the regression targets into $K = 10$ bins and computing the difference between inter-bin and intra-bin distances. Figures E.2 and E.3 show the results using cosine and Euclidean distances, respectively. Note that in the regression setting, a larger separation gap does not necessarily correspond to better predictive performance, as the discretization into bins introduces an approximation. Nevertheless, the separation gap generally increases with depth, consistent with our classification findings.

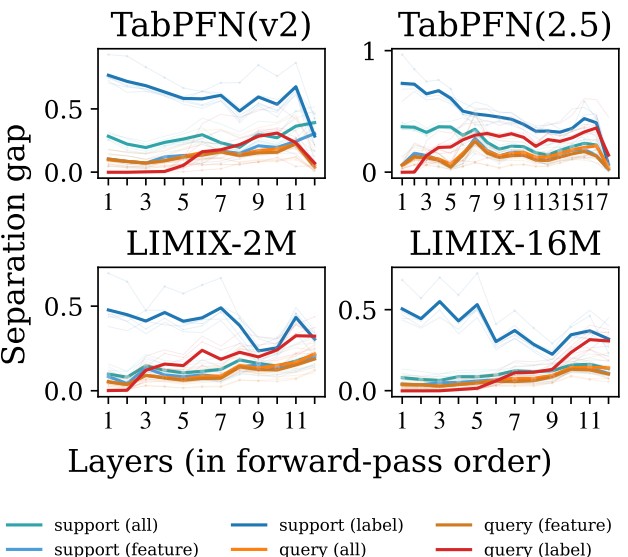

*Figure E.2.* Separation gap (cosine distance, after PCA) across layers for regression *TabArena* tasks. Bold lines indicate the average across tasks; thin lines represent individual datasets.

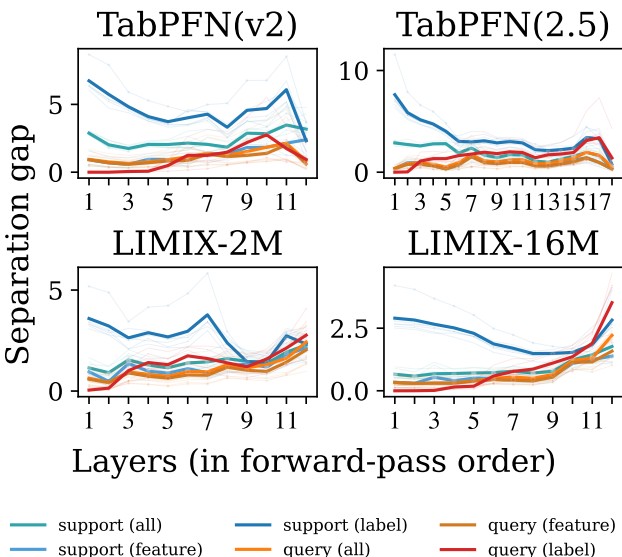

*Figure E.3.* Separation gap (Euclidean distance, after PCA) across layers for regression *TabArena* tasks.

## E.3. Probing Experiment

For regression tasks, we replace the logistic regression probing classifier with a random forest regressor, evaluating performance using normalized RMSE. Figure E.4 shows that the asymmetric transfer pattern observed in classification persists: a probe trained on layer $i$ generalizes better to later layers $j > i$ than the reverse. This confirms that each layer cumulatively enriches the representation by adding new features while preserving those from earlier layers.

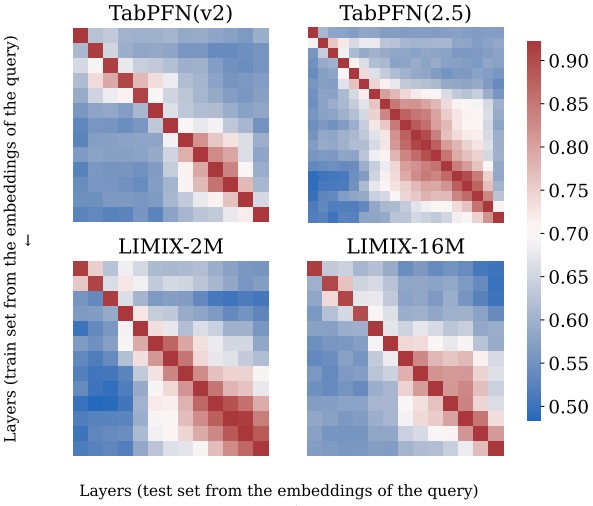

*Figure E.4.* Normalized RMSE for a random forest regressor used as a probing model, trained on embeddings at different layers for regression *TabArena* tasks. Lower values indicate better performance.

## E.4. Tabular Logit Lens

We adapt *TabICL* priors to regression tasks and pretrain individual decoders following the same protocol as in the classification setting (Appendix A.6). Figure E.5 reports performance at each layer using both negative RMSE (sensitive to calibration) and Spearman's rank correlation (invariant to calibration). The results indicate the same inference stages as discussed for classification: a rapid performance improvement in early-to-middle layers, followed by a calibration stage

in the final layers where rank correlation saturates while RMSE continues to improve. This suggests that the final layers primarily refine calibration rather than introducing fundamentally new predictive features, analogous to the prediction calibration stage identified in classification.

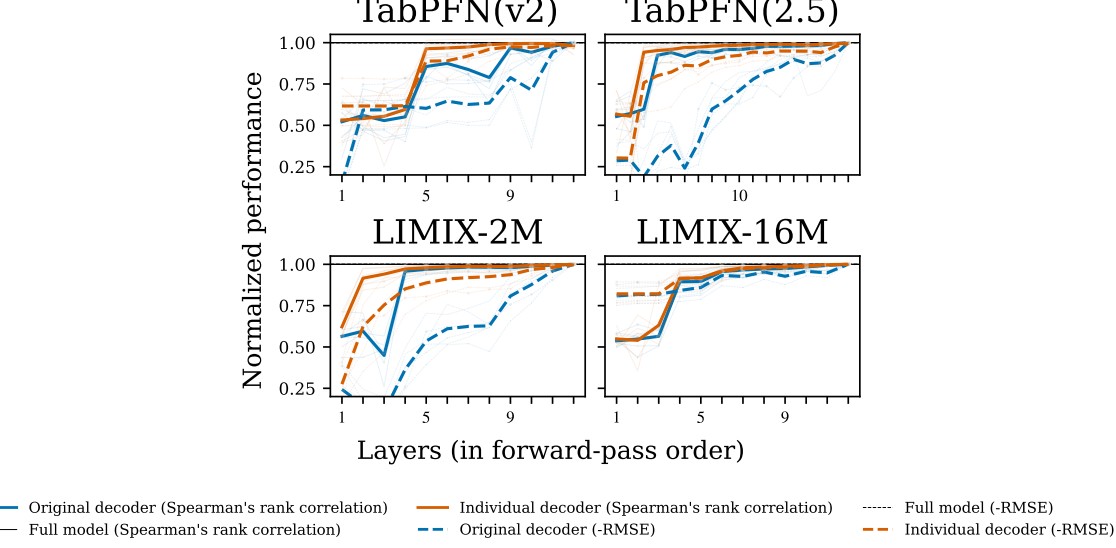

*Figure E.5.* Tabular logit lens results for regression *TabArena* tasks. We report Spearman's rank correlation (invariant to calibration) and negative RMSE (sensitive to calibration) across layers.

### E.5. Layer Ablation

Figure E.6 shows the effect of layer ablations on regression tasks. The results for the *LimiX-2M* and *LimiX-16M* families are consistent with the classification setting: early layers are the most sensitive, while middle and later layers are robust to ablation. For the *TabPFN* family, however, the final layer is also found to be important. A possible explanation is that the regression decoder must differentiate between up to 5,000 output bins, making the final layers more critical for the decoding stage. This pattern is analogous to residual sharpening observed in LLMs.

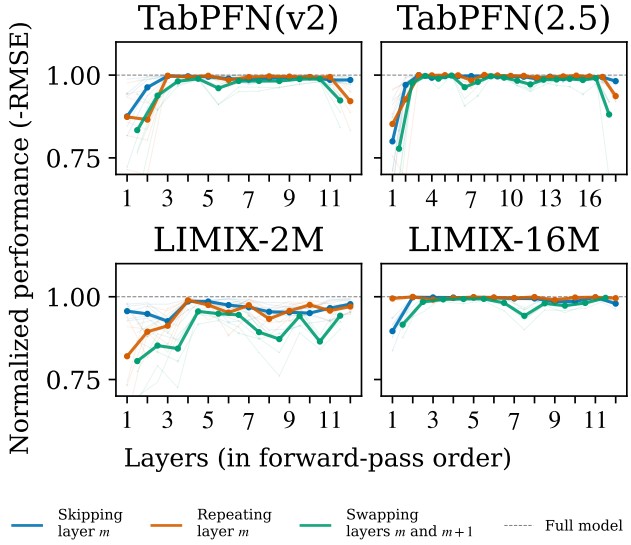

*Figure E.6.* Effect of layer ablation (skipping, repeating, swapping) on model performance for regression *TabArena* tasks.

## E.6. Self-Repair

Figure E.7 shows the self-repair analysis under layer skipping for regression tasks. Consistent with the classification setting, early-layer ablations (with the exception of *LimiX-2M*) cannot be recovered by subsequent layers. In contrast, skipping middle or later layers is followed by performance recovery, indicating that self-repair occurs and that these layers perform overlapping computations.

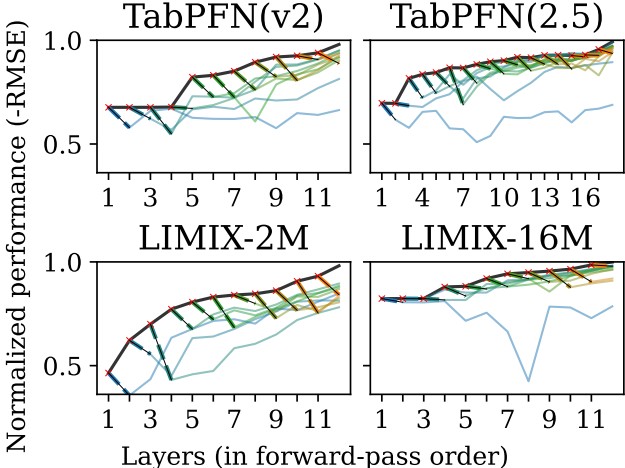

*Figure E.7.* Self-repair analysis under layer skipping for regression *TabArena* tasks. The solid black line shows intermediate performance without intervention. Colored lines (blue to orange, early to late) show intermediate performance after each layer ablation; a drop followed by recovery indicates self-repair.

## E.7. Is One Layer Enough?

We evaluate our proof-of-concept looped transformer on regression tasks. Figure E.8 shows the early-exit performance curves. The *nanoTabPFN$_{looped}$* achieves performance comparable to *nanoTabPFN$_{6l}$* across all datasets, while the single-layer *nanoTabPFN$_{1l}$* consistently underperforms. This confirms that the benefits of iterative layer reuse extend beyond classification to regression tasks.

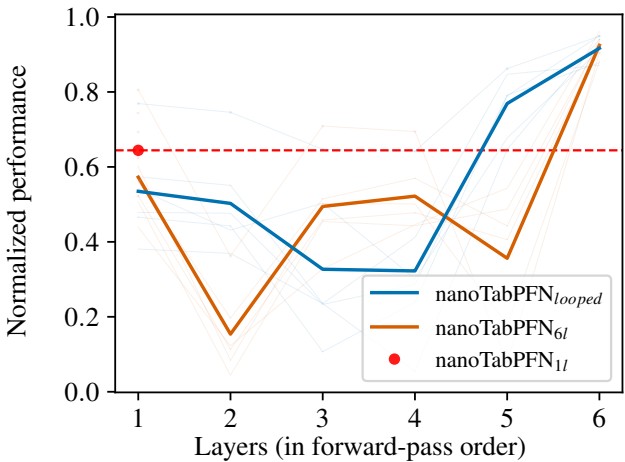

*Figure E.8.* Early-exit performance of *nanoTabPFN$_{6l}$*, *nanoTabPFN$_{1l}$*, and *nanoTabPFN$_{looped}$* on regression *TabArena* tasks.

