# OpenReview forum: "Is One Layer Enough? Understanding Inference Dynamics in Tabular Foundation Models"
_ICML.cc/2026/Conference — ICML 2026 regular_

### Official Review · Reviewer_GrrU · 2026-03-01

**Soundness:** 2
**Presentation:** 2
**Significance:** 2
**Originality:** 2
**Overall Recommendation:** 3
**Confidence:** 5

**Summary:**

The paper presents a mechanistic study of layerwise dynamics in six tabular foundation models. It adapts interpretability methods from large language models, such as structural interventions, representation analysis, and a modified logit lens, to understand how tabular models process data across layers. The authors identify distinct inference stages and observe substantial depthwise redundancy. Based on these observations, they introduce a looped single-layer model as a parameter-efficient alternative to multi-layer architectures.

**Compliance With Llm Reviewing Policy:**

Affirmed.

**Final Justification:**

While I appreciate the inclusion of the new regression experiments, some reservations about the broader claims remain. The reliance on a small-scale proof of concept leaves lingering questions about how well these findings might actually translate to larger, state-of-the-art architectures.

**Key Questions For Authors:**

- Can you provide empirical evidence that the looped single-layer architecture maintains competitive performance when scaled to the size of TabPFN(2.5) or LimiX-16M?

- How do the observed inference stages change when the models are applied to regression tasks instead of binary classification?

- The separation gap analysis relies on a 95 percent variance PCA projection. Have you investigated whether the discarded 5 percent contains non-linear features crucial for the final prediction calibration stage?

**Limitations:**

yes

**Strengths And Weaknesses:**

## Strengths:

- The comprehensive empirical evaluation across six distinct models provides a broad perspective on inference dynamics, preventing conclusions that overfit to a single architecture.

- The introduction of the tabular logit lens effectively solves the entropy collapse issues seen when applying standard logit lens techniques to tabular foundation models.

## Weaknesses:

- The looped transformer experiment is conducted solely on a miniature architecture rather than the main models studied (TabPFN, TabICL, LimiX). This mismatch undermines the core claim that a single layer is enough for state-of-the-art models.

- The evaluation is restricted entirely to binary classification tasks. The layerwise dynamics for multi-class classification or regression tasks remain completely unverified, severely limiting the scope of the findings.

---

> ### Author Rebuttal · Authors · 2026-03-31
>
> We thank the reviewer for their detailed feedback and clarify below how our experiments, analyses, and proposed methods address the concerns raised.
>
> ---
>
> ### Can you provide empirical evidence that the looped single-layer architecture maintains competitive performance when scaled to the size of TabPFN(2.5) or LimiX-16M?
>
> In this work, we **primarily focus on providing mechanistic evidence that depth in TFMs largely supports iterative refinement with overlapping computations**. Based on these insights, we demonstrate in a proof-of-concept experiment that a looped single-layer architecture can recover the performance of a six-layer model, establishing the feasibility of this design.  A large-scale evaluation of the looped architecture at the scale of TabPFN(2.5) or LimiX-16M is an important step for follow up work.
>
> Scaling this approach to models such as TabPFN(2.5) or LimiX-16M involves several non-trivial challenges beyond the scope of this paper: a) training  such large models is a challenging and compute-intensive task (as discussed in their respective publication), b) reproducing and ablating their design decisions requires access to (not yet available) data generating priors and training procedures and c) scaling a looped design also requires architectural modifications to ensure stable training, such as techniques to mitigate noisy gradients across repeated loop iterations [1] and time steps encoding[2].
>
> ---
>
> ### How do the observed inference stages change when the models are applied to regression tasks instead of binary classification?
>
> Thank you for raising this point.
>
> Only a subset of the studied TFMs support  regression. In the TabPFN family, regression is reformulated as a classification problem. In contrast, the LimiX family uses a shared transformer backbone with task-specific heads for regression or classification. Importantly, in both cases, the underlying transformer backbone that we analyze remains unchanged.
>
> Based on this, we expect that most of the observed inference stages, such as latent mapping, feature engineering and iterative refinement to be task-agnostic as they arise from the shared backbone, but we acknowledge that this has not been empirically evaluated. We will add this to the discussion.
>
> ---
>
> ### The separation gap analysis relies on a 95 percent variance PCA projection. Have you investigated whether the discarded 5 percent contains non-linear features crucial for the final prediction calibration stage?
>
> We report results without PCA in Appendix Figure B.2. Comparing Figures B.2 and B.3 shows that PCA improves the visibility of the signal, making fluctuations more pronounced. Nevertheless, our findings remain consistent without PCA, and the variation of the separation gap in the final layers is still observable.
>
> Additionally, it is important to clarify that the separation gap studies only the embeddings, without using a decoder and considering intermediate predictions. Studying the prediction calibration stage, by contrast, would require a fine-grained analysis of the differences between the original and individual decoders, which is beyond the scope of the separation gap analysis.
>
> ---
>
> ### … the layerwise dynamics for multi-class classification or regression tasks remain completely unverified….
>
> We agree that validating our findings beyond binary classification is important, however, we developed our analysis pipeline for binary classification tasks. This allows us to study layer dynamics in a clean, controlled, and consistent setup.
> Parts of our analysis pipeline are tailored to classification (such as the separation gap); however, the core phenomena we identify, such as iterative refinement, layerwise redundancy, and self-repair, arise from the transformer architecture and are not specific to the binary setting. See also our response to reviewer rvzJ and more details in [multiclass figures](https://anonymous.4open.science/r/is_one_layer_enough/multiclass.pdf), where we provide a subset of results for multi-class tasks showing initial evidence that our findings generalize beyond the binary setting.
> Finally, adapting the whole experimental framework to regression tasks requires additional methodological development (a prior to train individual decoders for regression and adaptation and development of metrics), which we leave for future work (explicitly listed as a limitation).
>
> ---
>
> ### References:
> [1] Wang, Shaowen, et al. "On the Residual Scaling of Looped Transformers: Stability and Transferability." Workshop on Latent {\&} Implicit Thinking {\textendash} Going Beyond CoT Reasoning.
>
> [2] Xu, Kevin, and Issei Sato. "On Expressive Power of Looped Transformers: Theoretical Analysis and Enhancement via Timestep Encoding." International Conference on Machine Learning. PMLR, 2025.

---

> > ### Author Rebuttal · Reviewer_GrrU · 2026-04-03
> >
> > I thank the authors for their response, but I am maintaining my score. While the paper presents an interesting hypothesis, the authors acknowledge that scaling their architecture to actual state-of-the-art models is beyond their current scope due to training and stability challenges, leaving the core claim validated only on a toy proof-of-concept. Furthermore, a foundational claim about tabular models is fundamentally incomplete without evaluating regression tasks, which the authors admit requires future methodological development. Finally, the chosen separation gap metric cannot account for the final prediction calibration stage, a critical phase of the inference dynamic remains unverified, meaning the empirical evidence is currently limited to support the paper's broad claims.

---

> > > ### Author Response · Authors · 2026-04-07
> > >
> > > We thank the reviewer for their elaboration on their concerns. We further clarify the points below to address their concerns:
> > >
> > > ---
> > >
> > > ### Regarding scaling
> > >
> > > As stated in the paper’s contributions (lines 66–94), the primary focus is on addressing RQ1 and RQ2. The proof-of-concept experiment serves as **supporting evidence for our findings rather than constituting the core claim.**
> > >
> > > Regarding scalability, prior work (Theorem 5.2 in [1]) establishes theoretical feasibility. Empirical scaling is orthogonal to our research questions as larger-scale settings introduce confounding architectural and optimization factors that complicate controlled analysis.
> > >
> > > For the controlled setting, we deliberately used nanoTabPFN to minimize such confounding factors.
> > > Notably, nanoTabPFN was explicitly designed to be small while maintaining strong performance, and we demonstrate that it can be made even smaller without losing performance. This is a remarkable result, as  **it highlights overlapping computation and supports our mechanistic insights.**
> > >
> > > ---
> > >
> > > ### Regarding the separation gap metric
> > > Yes, you are correct, the separation gap is not designed to measure calibration. However, **Tabular Logit Lens provides insights into when calibration happens**  (i.e., by analyzing how AUC and balanced accuracy evolve in the original and individual decoders). See Table C.1.
> > >
> > > ---
> > >
> > > ### Regarding missing regression tasks.
> > > We ran all experiments for regression and provide the results here: [regression](https://anonymous.4open.science/r/is_one_layer_enough/regression.pdf)
> > >
> > > Specifically, we adapted the experiments as follows:
> > > * We use all TFMs that support regression (LimiX-2M, LimiX-16M, TabPFN V2, and TabPFN 2.5).
> > > * We use negative RMSE as the primary performance metric and additionally report Spearman’s rank correlation, which is invariant to calibration.
> > > * The separation gap experiment is extended to regression by treating regression as a classification problem.
> > > * We adapt TabICL priors to regression tasks.
> > > * All six regression tasks in TabArena with fewer than 100 features and 10,000 samples are included, with experiments run using the repetitions and folds specified by TabArena.
> > > * We evaluate our proof-of-concept model on regression tasks.
> > >
> > > Overall, the results on regression tasks are consistent with and support our main findings.
> > >
> > > ---
> > >
> > > ### References:
> > > [1] Saunshi, Nikunj, et al. "Reasoning with Latent Thoughts: On the Power of Looped Transformers." The Thirteenth International Conference on Learning Representations.

---

### Official Review · Reviewer_8XJv · 2026-03-07

**Soundness:** 4
**Presentation:** 3
**Significance:** 4
**Originality:** 2
**Overall Recommendation:** 5
**Confidence:** 4

**Summary:**

This paper presents a large-scale mechanistic analysis of how tabular foundation models (TFMs) perform in-context learning across depth. The authors adapt a tuned-lens-style methodology to the tabular setting (training a per-layer “individual decoder”), and combine it with embedding similarity analysis (cosine/CKA), a class separation metric, linear probes, and structural interventions (skip/repeat/swap layers) to characterize inference dynamics in six state-of-the-art TFMs. The core findings suggest strong depth-wise redundancy and iterative refinement with overlapping computations, and motivate a proof-of-concept looped single-layer model that matches a six-layer baseline’s performance with about 20% of its parameters (though with similar compute).

**Compliance With Llm Reviewing Policy:**

Affirmed.

**Final Justification:**

The paper presents a methodologically sound mechanistic interpretability analysis of tabular foundation models, i.e. TabPFN and LimiX, offering valuable insights into their internal dynamics. I continue recommending acceptance.

**Key Questions For Authors:**

1. How sensitive are your layer-wise performance profiles to using a single TabICL prior for training all individual decoders?
2. Could you clarify exact query-train/test split ratios, fold alignment with ICL eval, and leakage safeguards in Section 3.3/ Appendix B.3. Please report per-dataset means±std for normalized AUC (Fig. 5) and quantitative cross-layer asymmetry.
3. For skip/repeat/swap interventions, how are LayerNorms (pre/post), positional/feature encodings, and attention masks treated in each architecture? Were any re-scaling or re-normalization steps applied to stabilize residual streams after structural edits?

**Limitations:**

Yes, the authors have discussed the limitations of their work.

**Strengths And Weaknesses:**

## Soundness
Strengths:
- Demonstrates a practical implication via a looped single-layer transformer that achieves near-parity with a deeper model while greatly reducing parameters.
- Methodology is sound and evaluations are comprehensive: representational similarity (cosine/CKA), a tailored separation-gap metric, linear probing, and structural interventions (skip/repeat/swap) with careful per-layer evaluation.
- Evaluation across six TFMs (including TabPFN 1 2 2.5, TabICL, and LimiX 2M 16M) is comprehensive.
- The self-repair analysis that measures recovery trajectories after ablations is insightful and goes beyond only end-of-model metrics.

Weaknesses:
- The per-layer “individual decoder” is trained using priors from TabICL, which may be mismatched to other TFMs and could bias conclusions about when sufficient information appears in hidden states if TabICL priors favor simpler representations decodable early, or inflate late-layer gains via better alignment. It is unclear whether similar conclusions hold if layer decoders are trained with each model’s own synthetic prior or with real-data supervision. Though it is understandable given the lack of access for model-specific code.
- The separation-gap metric, while intuitive, conflates geometry with class structure and requires PCA; Appendix B.2 shows raw vs PCA-smoothed gaps, but does not provide a deeper analysis of sensitivity or dataset confounds.
- Evaluation is restricted to binary classification and relatively small d,n regimes though this is acknowledged in the Limitations; it is not shown whether findings generalize to multi-class, regression, high-dimensional (>100 features), or very-large-N tables, settings where TFMs might show distinct behaviors.
- Lack of statistical testing for many claims (e.g., significance of differences in early-exit curves, ablation effects across datasets); the reported trends look robust but would benefit from quantification (e.g., confidence intervals on improvements, dataset-wise tests).

Overall, the technical work is solid and methodologically careful.

## Presentation
Strengths:
- Clear research questions, consistent narrative, and well-structured experiments whose roles are easy to understand.
- Figures and takeaways are concise and informative; limitations are acknowledged explicitly.

Weaknesses:
- No code is provided.
- More implementation detail is needed on how layer skipping/repeating/swapping is realized across different backbones (especially around LayerNorm, positional/feature encodings, and masking). Appendix B.5 covers results extensively but lacks implementation specifics.

## Significance
Strengths:
- Advances mechanistic understanding of a rapidly growing class of models (TFMs) used in practice.
- Offers concrete, actionable implications for TFM design and compression (e.g., layer tying/looping, potential early-exit, encoder strength).
- Introduces a “tabular tuned lens” by training per-layer decoders, showing early-layer representations already suffice for strong predictions but are misaligned with the original decoder.

## Originality
Strengths:
- The first broad, cross-model mechanistic study focused specifically on TFMs, bridging a gap with a literature dominated by LLM analyses.
- Identifies iterative refinement, block-like layer structures, and self-repair phenomena in TFMs, and connects these to architectural implications.

Weaknesses:
- Ye et al. [1] analyzes TabPFN v2’s internal dynamics and test-time strategies, it is closely related and could be referenced/discussed in the main text.

[1] Ye, H.-J., Liu, S.-Y., & Chao, W.-L. (2025). A closer look at TabPFN v2: Understanding its strengths and extending its capabilities. In The Thirty-ninth Annual Conference on Neural Information Processing Systems.

---

> ### Author Rebuttal · Authors · 2026-03-31
>
> We thank the reviewer for their thorough evaluation and positive feedback, and we address their points regarding decoder priors, metric sensitivity, generalization, and implementation details below.
>
> ---
>
> ### How sensitive are your layer-wise performance profiles to using a single TabICL prior for training all individual decoders?
>
> Thank you for raising this point.  Using an appropriate prior is important to train the per-layer decoders. In the tabular logit lens experiment (exp. 4), the individual decoder and the original final-layer decoder achieve very similar performance at the final layer, indicating that the TabICL prior provided a sufficiently good alignment for our analysis. Furthermore, the decoder in each TFM is a small MLP (typically <1–4% of total parameters; see Appendix A.2), and we continue training from the original weights rather than starting from scratch. For these reasons, the sensitivity to a specific choice of prior and the stochasticity of the training process are limited.
>
>
> ---
>
> ### Could you clarify exact query-train/test split ratios, fold alignment with ICL eval, and leakage safeguards ….
>
> Regarding Leakage Safeguards in Section 3.3: Experiment 2 shows that the embeddings derived from the support set encode label information. Training a probing classifier directly on the support set embeddings would contain label information unavailable at test time.
>
> To avoid this, we split the original training data into two disjoint subsets, resulting in $D_{train} = D_{train1} \cup D_{train2}$ and the original $D_{test}$. $D_{train1}$ is used as the support set for in-context learning (ICL), and $D_{train2} \cup D_{test} $ is used as the unlabelled query set. We train the probing classifier on embeddings of $D_{train2}$ and evaluate its generalization on embeddings of $D_{test}$.
> With this setup, the test data used to evaluate the probing classifier is exactly the same as the original test set, and performance is directly comparable. We will extend our explanation of this procedure in Section 3.3, lines 255-262).
>
> We will also report per-dataset AUC and a quantitative version of Figure 5.
>
> ---
>
> ### For skip/repeat/swap interventions, how are LayerNorms (pre/post), positional/feature encodings, and attention masks treated in each architecture?...
>
> Thank you for your question. We apply skip, repeat, and swap interventions strictly at the level of transformer blocks; throughout, a “layer” refers to a single transformer block. Most importantly, for all interventions, the internal structure of each block remains unchanged and we do not modify the in- or outputs to the transformer stack or after structural edits.
>
> In particular,
>   * the pre- and post-LayerNorm operations are preserved as part of the block and move together with it when layers are skipped, repeated, or swapped
> positional and feature encodings are applied before the transformer blocks and are therefore unaffected by these interventions,
>    * attention masks remain fixed across all experiments,
>   * and, finally, we do not apply any additional re-scaling or re-normalization after structural edits, as the LayerNorm operations within each transformer block are sufficient to maintain stability.
>
> ---
>
> ### No code is provided. More implementation detail
> We will make our code available. The rebuttal guidelines seem not to allow us to provide a link to an anonymized repo (“and links may only be used for figures (including tables) and captions that describe the figure (no additional text)”). If you’re interested in further implementation details, we’re happy to clarify.
>
> ---
>
> ###  Related work Ye et al. (2025)
> We agree that this paper is related to our work. We already discussed Ye et al. (2025) in the related work. Still, we will expand the discussion in the main text to better highlight their contributions and clarify the differences to our approach.
>
> ---
>
> ### Evaluation is restricted to binary classification
> We add multiclass results and more detail in [multiclass figures](https://anonymous.4open.science/r/is_one_layer_enough/multiclass.pdf) (see also our response to reviewer rvzJ).

---

> > ### Author Rebuttal · Reviewer_8XJv · 2026-04-01
> >
> > I thank the authors for their response. I continue recommending acceptance.

---

### Official Review · Reviewer_rvzJ · 2026-03-11

**Soundness:** 2
**Presentation:** 4
**Significance:** 3
**Originality:** 4
**Overall Recommendation:** 4
**Confidence:** 5

**Summary:**

This study presents a pioneering mechanistic interpretability analysis of Tabular Foundation Models (TFMs). By examining the layerwise dynamics of 6 models, the authors identify a consistent pattern of depthwise redundancy and delineate the inference process into three functional stages: de-tokenization, feature refinement, and prediction ensembling. The work culminates in the proposal of a "looped single-layer model," demonstrating that iterative computation with a significantly reduced parameter set can achieve performance parity with deeper, conventional architectures on standard benchmarks.

**Compliance With Llm Reviewing Policy:**

Affirmed.

**Key Questions For Authors:**

I. Generalizability across Task Types

The empirical evidence presented is based exclusively on binary classification datasets. Compared to multiclass classification or regression, binary tasks involve a relatively straightforward projection in the logit space.

Question: Does the observed iterative refinement mechanism hold when the model must navigate the more complex decision boundaries of multiclass tasks or the continuous output spaces of regression? The authors are encouraged to provide evidence from multiclass datasets to ensure that the "one layer is enough" hypothesis is not a byproduct of task simplicity.

II. Relationship between Data Complexity and Computational Depth

The depthwise redundancy identified in this study may be a function of the intrinsic complexity of the datasets utilized, such as those in the PMLBmini collection.

Question: There appears to be a gap in the analysis regarding how data complexity dictates the required "effective depth" of a model. The authors should investigate the boundary conditions under which this redundancy might vanish. For instance, are there datasets with high-order feature interactions or highly non-linear manifolds that necessitate the unique, non-redundant transformations provided by a deep architecture? Providing a more rigorous analysis of the coupling between task difficulty and the stability of the "inference blocks" would significantly strengthen the theoretical contributions of this work.

III. Insufficient Baselines for the Looped Model

The authors demonstrate that a single layer looped five times can match the performance of a six-layer model, suggesting that iteration is the key to efficiency.

Question: The current experimental design lacks a critical baseline: a non-looped model of equivalent parameter scale. Without comparing the looped model against a standard shallow architecture with a similar number of parameters, it remains unclear whether the performance stems from the iterative mechanism itself or simply from the fact that the tasks do not require high parameter density. To isolate the value of the "looping" architecture, a comparison against an appropriately scaled non-looped counterpart is necessary.

**Limitations:**

See above.

**Strengths And Weaknesses:**

The manuscript introduces a valuable perspective by applying mechanistic interpretability to the relatively opaque field of tabular foundation models. The observation that these models exhibit substantial computational redundancy is both insightful and highly intuitive. From the perspective of k-nearest neighbors (k-NN) or Bayesian posterior sampling, tabular decision boundaries are frequently determined by the local distribution of support samples. Consequently, the traditional deep Transformer architecture may indeed be over-parameterized for these tasks. By providing empirical evidence for this "iterative refinement" mechanism, the authors offer a promising direction for developing more efficient, specialized architectures for tabular data.

---

> ### Author Rebuttal · Authors · 2026-03-31
>
> We thank the reviewer for recognizing the novelty and insightful mechanistic interpretation of TFMs, and for highlighting the importance of our findings on depthwise redundancy and iterative refinement. We address your questions in the following.
>
> ---
>
> ### … does the mechanism persist in multiclass and regression settings? Is “one layer is enough” robust to multiclass, or just a byproduct of simple decision boundaries?
>
> We focus on binary classification because these tasks represent the majority of commonly used community benchmark tasks.  Also, for solving a tabular task, including binary, multi-class, and regression, the model needs to capture key phenomena such as feature interaction, distributional patterns, and local decision boundaries; thus, being binary does not make the problem inherently simpler and does not require fundamentally different inference mechanisms. Insights gained here can later guide extensions to multiclass or regression tasks (see our response to reviewer GrrU), which largely involve scaling and specialization rather than fundamentally different mechanisms.
>
> We evaluated our POC models (Section 5) on $6$ of the $8$ multiclass datasets of TabArena (excluding two datasets due to memory constraints). The tables below show that  nanoTabPFN$\_{looped}$  outperforms nanoTabPFN$\_{1l}$, with both models having the same number of parameters. Moreover, nanoTabPFN$\_{looped}$ achieves similar performance to nanoTabPFN$\_{6l}$. This is consistent with our findings on binary classification, further validating our approach (see [multiclass figures](https://anonymous.4open.science/r/is_one_layer_enough/multiclass.pdf) ). We will add this result to the paper.
>
> ### Balanced Accuracy (higher is better)
> |   task id | nanoTabPFN$_{looped}$   | nanoTabPFN$_{6l}$   | nanoTabPFN$_{1l}$   |
> |----------:|:------------------------|:--------------------|:--------------------|
> |    363614 | 0.81 ± 0.06             | 0.78 ± 0.02         | 0.36 ± 0.02         |
> |    363685 | 0.68 ± 0.03             | 0.70 ± 0.02         | 0.56 ± 0.02         |
> |    363702 | 0.91 ± 0.01             | 0.91 ± 0.01         | 0.33 ± 0.00         |
> |    363704 | 0.66 ± 0.01             | 0.68 ± 0.01         | 0.49 ± 0.01         |
> |    363707 | 0.63 ± 0.01             | 0.63 ± 0.01         | 0.50 ± 0.01         |
> |    363711 | 0.20 ± 0.04             | 0.18 ± 0.05         | 0.13 ± 0.01         |
>
> ---
>
> ### … data complexity dictates the required "effective depth" of a model…
>
> We thank the reviewer for this insightful comment. We agree that understanding when “effective depth” becomes necessary is important. Tabular data is inherently heterogeneous, often comprising a mix of numerical, categorical, ordinal, and sometimes missing or sparse features [1]. As a result, heterogeneity makes it difficult to define a unified notion of tabular task complexity. Unlike domains such as vision or language, where structure (e.g., spatial or sequential locality) provides a natural proxy, tabular datasets exhibit irregular and dataset-specific interaction patterns.
>
> While we observe some variation across datasets, the overall trends are consistent. In preliminary studies, we explored correlations between our observations and basic dataset characteristics (e.g., number of samples, number of features, and number of categorical features), but did not observe strong or consistent patterns.  A more systematic study would likely require carefully designed benchmark tasks that vary task complexity and other characteristics in a controlled manner. We agree that this is an interesting direction for future work and will clarify this point in the revised version.
>
>
> ---
>
> ### ..lacks a critical baseline: a non-looped model of equivalent parameter …
>
> We apologize for the lack of clarity.
>
> **This baseline is already included in our experiments, called nanoTabPFN$\_{1l}$ (see Figure 9; red dot). nanoTabPFN$\_{1l}$ is a 1-layer transformer with the same number of parameters as the looped transformer.**
>
> To clarify the comparison: in our proof-of-concept experiments, for the looped transformer, a single transformer block is iterated six times. However,  both models share input encoders and output decoders (see Tables A.7 and A.8). As a result, the total parameter count of the looped version is approximately five times smaller.
>
> This controlled comparison suggests that the observed performance gains are not merely due to parameter count, but are instead attributable to the iterative (looping) mechanism.
>
> We will revise the paper to make this comparison more explicit.
>
> ---
> ### References:
> [1] Grinsztajn, Léo, Edouard Oyallon, and Gaël Varoquaux. "Why do tree-based models still outperform deep learning on typical tabular data?." Advances in neural information processing systems (2022).

---

> > ### Author Rebuttal · Reviewer_rvzJ · 2026-04-06
> >
> > This is an insightful paper. While the discussion on the relationship between data complexity and computational depth did not entirely address my initial concerns, I acknowledge that this is a multifaceted issue that a single study cannot comprehensively solve.
> >
> > Although the depth of analysis remains somewhat limited, I think that this work satisfies the criteria for ICML. I will retain my positive rating. Additionally, the regression and multi-class classification experiments should be incorporated into the camera-ready version, regardless of their performance outcomes. Drawing upon my experience with this field, I find regression tasks to be inherently more challenging.

---

> > > ### Author Response · Authors · 2026-04-07
> > >
> > > We thank the reviewer for their positive feedback.
> > >
> > > In response to the suggestion, we have conducted additional experiments on regression tasks and provide the corresponding results (see [regression figures](https://anonymous.4open.science/r/is_one_layer_enough/regression.pdf) and our response to reviewer GrrU). We will add these results to the camera-ready version.

---

### Decision · Program_Chairs · 2026-04-30

**Decision:**

Accept (regular)

**Comment:**

This paper executes a mechanistic interpretability analysis of Tabular Foundation Models to explain the latent space dynamics of these models across depth. The authors identify a depth-based redundancy that is distinct from large language models. Reviewers appreciated the contributions overall, and the main objection after discussion was the limitation of the analysis to smaller models. While this is certainly a limitation, the authors are transparent about this, and this is easily outweighed by the novelty of the problem and the depth of the empirical evaluation. This work presents a useful first step in this direction.